# Daily caloric restriction limits tumor growth more effectively than caloric cycling regardless of dietary composition

Laura C. D. Pomatto-Watson[1], Monica Bodogai[2], Oye Bosompra[1,8], Jonathan Kato[1,8], Sarah Wong[1,8], Melissa Carpenter[1,8], Eleonora Duregon[1,8], Dolly Chowdhury[1], Priya Krishna[1], Sandy Ng [1], Emeline Ragonnaud[2], Roberto Salgado [3,4], Paula Gonzalez Ericsson [5], Alberto Diaz-Ruiz [6], Michel Bernier [1], Nathan L. Price[1], Arya Biragyn[2], Valter D. Longo [7] & Rafael de Cabo [1✉]

Cancer incidence increases with age and is a leading cause of death. Caloric restriction (CR) confers benefits on health and survival and delays cancer. However, due to CR's stringency, dietary alternatives offering the same cancer protection have become increasingly attractive. Short cycles of a plant-based diet designed to mimic fasting (FMD) are protective against tumorigenesis without the chronic restriction of calories. Yet, it is unclear whether the fasting time, level of dietary restriction, or nutrient composition is the primary driver behind cancer protection. Using a breast cancer model in mice, we compare the potency of daily CR to that of periodic caloric cycling on FMD or an isocaloric standard laboratory chow against primary tumor growth and metastatic burden. Here, we report that daily CR provides greater protection against tumor growth and metastasis to the lung, which may be in part due to the unique immune signature observed with daily CR.

[1] Experimental Gerontology Section, Translational Gerontology Branch, National Institute on Aging, National Institutes of Health, Baltimore, MD 21224, USA. [2] Immunoregulation Section (IS), Laboratory of Molecular Biology and Immunology, National Institute on Aging, National Institutes of Health, Baltimore, MD 21224, USA. [3] Department of Pathology, GZA-ZNA Hospitals, Antwerp, Belgium. [4] Peter Mac Callum Cancer Centre, Melbourne, Australia. [5] Breast Cancer Research Program, Vanderbilt University Medical Center, Nashville, TN 37232, USA. [6] Nutritional Interventions Group, Precision Nutrition and Aging Institute, IMDEA Food, Crta. de Canto Blanco n° 8, E–28049 Madrid, Spain. [7] Longevity Institute, School of Gerontology, and Department of Biological Sciences, University of Southern California, Los Angeles, CA 90089, USA. [8] These authors contributed equally: Oye Bosompra, Jonathan Kato, Sarah Wong, Melissa Carpenter, Eleonora Duregon. ✉email: decabora@mail.nih.gov

Caloric restriction (CR) is the most effective intervention to reduce the incidence and progression of most spontaneous and induced cancers. Due to the stringency of CR and its associated limitations, including low compliance among study participants and impaired wound healing[1], alternative dietary interventions are increasingly being considered as viable strategies to combat cancer. These approaches that include modifications of feeding frequency, diet composition, and or length of fasting often recapitulate CR-mediated benefits by conferring cancer protection[2]. Much of the improvement from daily CR is attributed to a sustained reduction in overall caloric intake and periods of prolonged fasting[2], a frequently overlooked variable that contributes not only to the activation of cellular maintenance and repair pathways, but also to extending health and survival[3]. Most CR regimens utilize a once-a-day feeding protocol which, depending on the level of restriction, can lead to a fasting period of up to 22 h[2,4].

Earlier work has shown that short periods of very low caloric intake, including either periods of short-term fasting (2–4 days) or dietary manipulation of specific macronutrients, can be effective at delaying primary tumor growth[4,5]. Conversely, excess consumption of animal-derived protein is linked with increased cancer risk and all-cause mortality[6,7]. Different forms of intermittent fasting (IF) and time-restricted feeding (TRF)[2,3] are broadly characterized by cyclical periods of low caloric intake or complete fasting interspersed between periods of ad libitum (AL) feeding. IF and TRF result in a dramatic reduction in tumor growth[8,9] and have garnered traction both as an adjuvant to chemotherapy and as a tool for cancer prevention with promising translational applications[10,11].

Periods of prolonged fasting result in decreased circulating blood glucose and IGF-1 signaling in target tissues[10], thus dampening tumor growth. Under low glucose conditions, normal cells undergo growth arrest, whereas malignant cells no longer respond to these conditions and maintain uncontrolled cell division. Consequently, a striking difference in the response of normal and cancerous cells to chemotherapy under fasting conditions has emerged, whereby normal cells, but not cancer cells, are protected from the cell-killing actions of anticancer drugs[12]. Therefore, much interest has centered on developing dietary approaches that recapitulate the selective targeting of cancer cells without the burden of CR. A plant-based diet, recently designed to mimic the physiological response to fasting ('fasting mimicking diet', FMD), was developed to minimize the burden of fasting while providing adequate micronutrients (vitamins, minerals, etc.), and to elicit beneficial improvements in metabolic parameters[13]. Periodic cycles (4-day cycle twice a month) of FMD followed by AL feeding promote health span in mice and humans[13] and confer protection against primary tumorigenesis, with or without chemotherapy[14–17]. This approach was also demonstrated to lower toxicity to chemotherapy in clinical trials[18].

Although these findings highlight the important role dietary interventions play in regulating tumor growth, it remains unclear whether the anti-tumorigenic benefits of CR, IF, and FMD are mediated by the salutary effect of diet composition, reduction in caloric intake, duration of fasting, or a combination of all these elements. In one study, Brandhorst et al. showed that 3-day cycles of 50% CR combined with chemotherapy did not delay tumor progression in a 4T1 breast cancer mouse model[19]. In contrast, severe protein restriction in an otherwise isocaloric diet was shown to slow down the progression of melanoma, but not breast cancer or glioma[6,19].

In this work, we assess the relative impact of diet composition vs. low caloric intake in delaying tumor growth in the 4T1 breast cancer mouse model. Tumor-bearing mice are subjected to two 4:10 feeding cycles, with 4 days of severe reduction in caloric intake in animals fed either FMD or standard laboratory chow

(AIN-93G), followed by 10 days of AL feeding with AIN-93G. Using this approach, we have been able to evaluate the extent to which diet composition impacts the response to 4:10 feeding cycles and whether this approach is as effective as daily CR at delaying tumorigenesis. Our findings show that compared to daily CR, 4:10 feeding cycles are less effective and fail to protect against lung metastases, regardless of diet composition or treatment initiation period (pre- or post-4T1 injection). Importantly, daily CR elicits a unique signature of immune activation by significantly reducing the number of tumor-promoting immune cells (CD11b+Gr1+), while upregulating tumor-fighting (CD8+ and CD4+) immune cells in peripheral tissues. These findings suggest that the duration and degree of CR are the most critical factors in determining protection against cancer progression in the 4T1 murine breast cancer model.

## Results

**Low-calorie cycles slow tumor growth independent of diet composition.** The impact of diet composition and 4:10 feeding cycles on the growth rate of triple-negative breast cancer (TNBC) was studied in 16-week-old female BALB/cJ mice implanted with syngeneic and highly metastatic murine 4T1 cancer cells in the mammary gland. The responses of 4:10 cycles of FMD vs. an isocaloric standard laboratory chow (AIN-93G), also known as 'low caloric cycling diet' (LCC), were compared. During the four days of severe low-calorie intake, FMD and LCC mice were exposed to a 50%:70%:70%:70% reduction in daily calories, followed by 10 days of AL feeding with AIN-93G diet (Fig. 1a, Supplementary Fig. 1a). One week after injection of 4T1 tumor cells, mice that were subjected to two 4:10 cycles of FMD or LCC (Fig. 1b) showed similar declines in tumor growth rate (Fig. 1c) and tumor area (Fig. 1d) compared to AL controls. FMD and LCC mice had identical body weight (Fig. 1e, Supplementary Fig. 1b) and food consumption (Fig. 1g) trajectories during the two cycles, with an overall decrease both in the average body weight (Fig. 1f) and caloric intake (Fig. 1h) across the 28-day period. These results suggest that cycles of very low-calorie intake, rather than diet composition per se, are the main driver behind delayed tumorigenesis.

**Therapeutic CR intervention slows tumor growth more than caloric cycling.** Prior studies have demonstrated the effectiveness of CR in delaying breast tumor growth[20,21]. We sought to compare whether 4:10 cycles of FMD and LCC are as effective as daily CR at delaying primary tumor growth. The amount of daily CR was determined based upon the average caloric intake of each 4:10 cycle of FMD/LCC, which was ~20% lower than AL. Seven days after tumor implantation, a CR group that underwent daily 20% CR on the AIN-93G diet (CR post-4T1) was introduced at the same time as the caloric cycling groups (FMD and LCC) (depicted in Fig. 2a). All interventions resulted in a significant reduction in body weight over the course of the experiment (Fig. 2b, Supplementary Fig. 2a), with FMD and LCC mice achieving a ~20% reduction in average caloric intake, similar to the amount of restriction in the daily CR group (Fig. 2c).

At the time of sacrifice (third day of refeeding on the second cycle of FMD/LCC, which amounts to 21 days on CR), various metabolic markers were assessed. Glucose levels showed a downward trend in CR mice (Fig. 2d), while insulin levels and HOMA2-IR values were elevated in the FMD and LCC groups (Fig. 2e, f). A significant reduction in IGF-1 levels occurred in CR and FMD mice, with a similar trend in the LCC group (Fig. 2g). The divergence in metabolic outcomes highlights the differences between daily CR vs. short cycles of CR. Indeed, CR mice

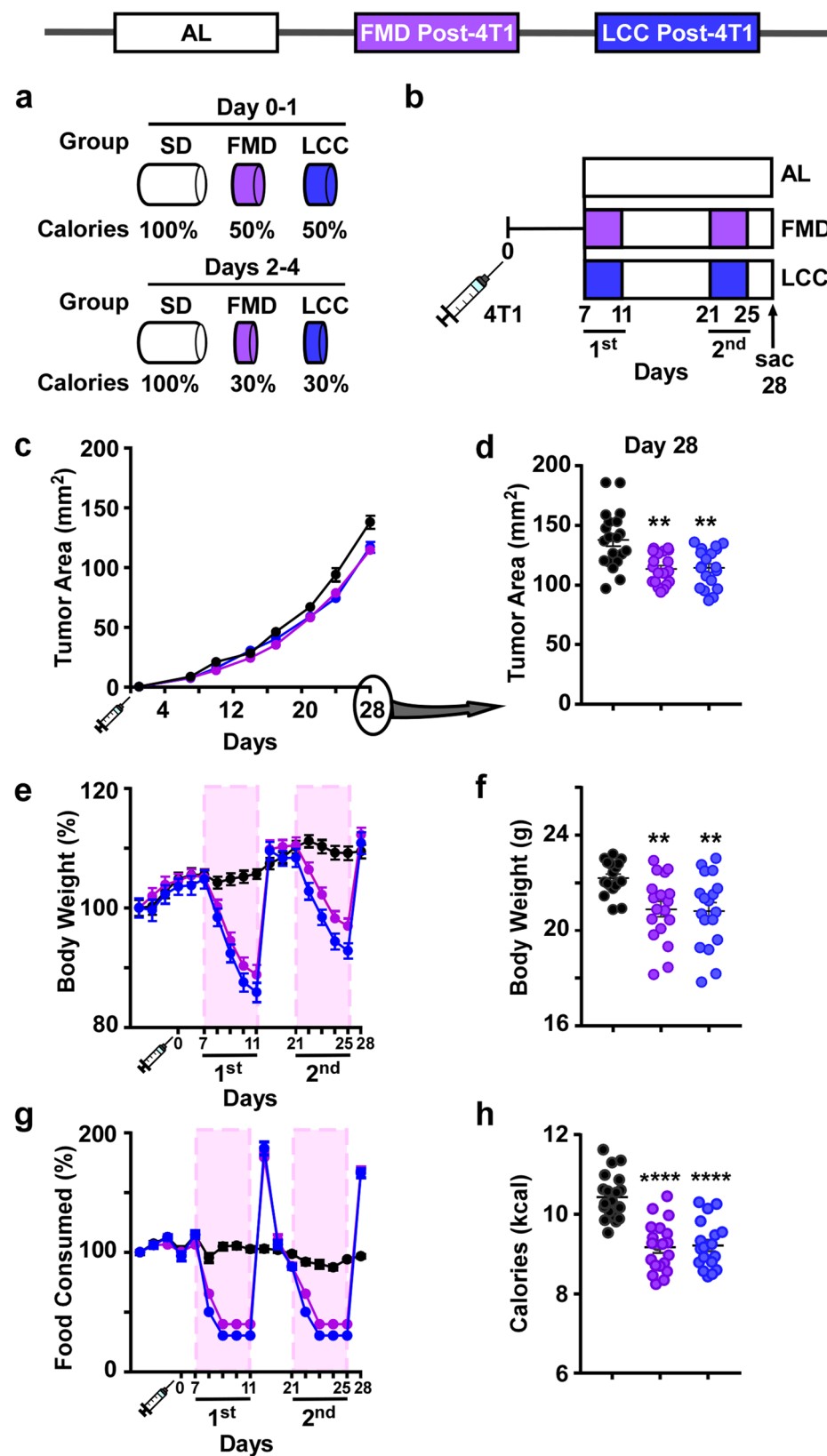

remained in a state of restricted calories up until the time of sacrifice, while FMD and LCC mice were sacrificed three days into the second refeeding period. This built-in caveat may explain the metabolic marker shifts observed in our study. The collection of samples in the fed vs. fasted state results in dramatic shifts in metabolic markers[13,17] and hepatic gene expression[22]. Therefore,

the increased liver mass in FMD and LCC mice (Fig. 2h) and the spike in select metabolic markers at the beginning of the refeeding period may be an indicator of the compensatory overfeeding FMD and LCC mice engage in response to the 4-day period of very low caloric intake (70% restriction). Assessment of liver protein, triglyceride, and glycogen content showed only a

**Fig. 1 Caloric cycling provides protection against primary tumor growth. a** Percent reduction in caloric intake during the 'ON' period of the fasting mimicking diet (FMD) and very low-calorie cycle (LCC) regimens (Days 0–4) compared to ad libitum (AL) feeding on standard laboratory chow (SD). **b** Experimental layout. 7 days after injection with 4T1 breast cancer cells (10[6] cells/mL) 16-week-old BALB/cJ females were subjected to caloric cycling (FMD or LCC) or maintained on AL. A second round of caloric cycling was performed on days 21–25, after which tissues were collected at day 28. **c** Growth rate of primary tumors. **d** Tumor area on day 28. **e** Percent change in body weight trajectories. **f** Average body weight during the course of the study. **g** Percent change in food consumption from baseline. **h** Average caloric intake. Most of the data are represented as scatter plots with mean values ± SEM. One-way ANOVA with Tukey post hoc analysis was used to determine statistical significance with **$p < 0.01$, ****$p < 0.0001$ compared to AL. **c–h** AL, $n = 20$; FMD, $n = 19$; LCC, $n = 19$ mice per treatment group. Source data are provided as a Source Data file.

significant decrease in total protein that was evident in all three dietary paradigms (Supplementary Fig. 2b–d).

Every intervention tested has resulted in a significant delay in tumor growth (Fig. 2i, j) and reduction in tumor mass (Fig. 2k), especially in mice fed CR each day vs. AL-fed controls. Thus, our findings confirm a century-old view of daily reduction in caloric intake hindering tumor growth rate. An additional treatment arm was also tested to further ascertain the impact of dietary differences. Mice that underwent daily CR on the FMD diet (CR-FMD) exhibited a severe loss in body weight, even though they were isocaloric to CR post-4T1 animals, making the CR-FMD regimen unsuitable for further investigation. Because the FMD diet was not designed for sustained use, these findings indicate that it may lack the adequate amount of key nutrients needed for long-term survival.

The positive association between tumor progression and splenic accumulation of tumor-promoting immune cells (splenomegaly)[7] led us to measure spleen mass, which was markedly lower in mice on daily CR and cycles of FMD or LCC (Fig. 2l). We surmise that the decrease in tumor growth could be stemming, at least in part, from a drop in the number of tumor-promoting immune regulatory cells. None of the dietary interventions were capable at reducing tumor viability as assessed in Hematoxylin & Eosin (H&E)-stained tumor whole slide images (Fig. 2m). These outcomes are important because of the growing interest in pairing dietary regimens with current standard cancer treatments, including chemotherapy[23]. Thus, the reduction in calorie intake and/or extended periods of fasting appear to be the key drivers of tumor growth decline, regardless of diet composition. However, daily CR was more effective than 4:10 cycles of FMD and LCC at delaying 4T1 tumorigenesis in vivo, even though they underwent a similar degree of calorie restriction.

**Preventive and therapeutic effect of CR in delaying tumor growth.** To ascertain whether prophylactic intervention of CR prior to 4T1 implantation has a greater improvement in delaying 4T1 primary tumor growth compared to therapeutic CR intervention, an additional treatment arm was included in the study presented in Fig. 2 (experimental design shown in Supplementary Fig. 3a). The findings from Fig. 2 (depicted as gray boxes in Supplementary Fig. 3) were included to better compare across treatment groups. Prior to 4T1 cell injection, mice in the CR group underwent a stepwise reduction in caloric intake in order to minimize metabolic stress (Supplementary Fig. 3a). CR intervention initiated pre- and post-4T1 cell implantation resulted in a significant reduction in average body weight (Supplementary Fig. 3b) and at three time points (Supplementary Fig. 3c) throughout the experiment. As anticipated, average caloric intake was also significantly lower compared to the AL control group (Supplementary Fig. 3d). Introduction of CR prior to 4T1 cell implantation led to a significant reduction in blood glucose levels (Supplementary Fig. 3e), which may be indicative of a metabolic adaptation period that preceded 4T1 cell injection (42 days versus 21 days of CR). However, insulin and HOMA2-IR index were not significantly impacted whether CR was initiated pre- and post-4T1

cell injection vs. AL controls (Supplementary Fig. 3f, g), whereas IGF-1 levels were significantly lower with CR intervention (Supplementary Fig. 3h). Post-4T1 CR mice had significantly lower liver weight (Supplementary Fig. 3i), and significant decrease in total liver protein was observed in response to CR (pre- and post-4T1 cell implantation) (Supplementary Fig. 3j), without differences in triglyceride or glycogen levels compared to AL controls (Supplementary Fig. 3k, l).

Irrespective of whether it is initiated pre- or post-4T1 cell implantation, CR was equally effective at delaying primary tumor growth (Supplementary Fig. 3m, n) and reducing the overall mass of the primary tumor (Supplementary Fig. 3o) and spleen (Supplementary Fig. 3p). However, primary tumor viability quantified in H&E-stained tumor sections was not reduced by daily CR in both groups of tumor-bearing mice (Supplementary Fig. 3q).

**Preventive and therapeutic effect of CR in reducing metastatic burden.** Next, we evaluated the impact of these dietary interventions on metastatic spread of 4T1 cells, as metastatic invasion of distal sites, particularly the lungs, is the leading cause of mortality in breast cancer patients[24]. Orthotopically implanted 4T1 cancer cells in the mammary gland of BALB/cJ readily metastasize into various sites, including the lungs[25], providing a useful in vivo model to assess whether daily CR or FMD and LCC cycling regimens can delay the development and progression of lung metastases. After 28 days of 4T1 tumor challenge, mice on daily CR (pre- or post-4T1 implantation) had a marked reduction in metastatic nodules in the lungs (Fig. 3a, b), a finding consistent with the recent report by De Lorenzo et al.[20]. Unexpectedly, FMD and LCC regimens did not confer such benefits, as metastatic foci in the lungs of FMD and LCC mice were not significantly different compared to AL controls (Fig. 3a, b). Mice on CR (pre- or post-4T1 implantation), but not on FMD or LCC, also had less visible large pulmonary metastases (>1.5 mm in size) (Fig. 3c), which are indicative of more advanced metastatic growth[20,26]. Only the daily CR (pre- and post-4T1 implantation) groups exhibited a significant reduction in metastatic lesions in H&E-stained lung sections (Fig. 3d). Overall, we conclude that CR exerts prophylactic and therapeutic effects by conferring anti-cancer protection against the growth and spread of a murine model for TNBC.

**Prophylactic CR slows tumor growth better than caloric cycling when both were begun concurrently.** To assess whether the benefits gained from the prophylactic administration of daily CR could be recapitulated in the caloric cycling groups, 16-week-old Balb/cJ female mice underwent two cycles of FMD or LCC prior to 4T1 injection (experimental design presented in Fig. 4a). Compared to the AL control group and mice on prophylactic CR, the caloric cycling groups showed a similar decrease in average body weight (Fig. 4b) and caloric intake (Fig. 4c), with cyclical fluctuations evident in the FMD and LCC groups (Supplementary Fig. 4a, b).

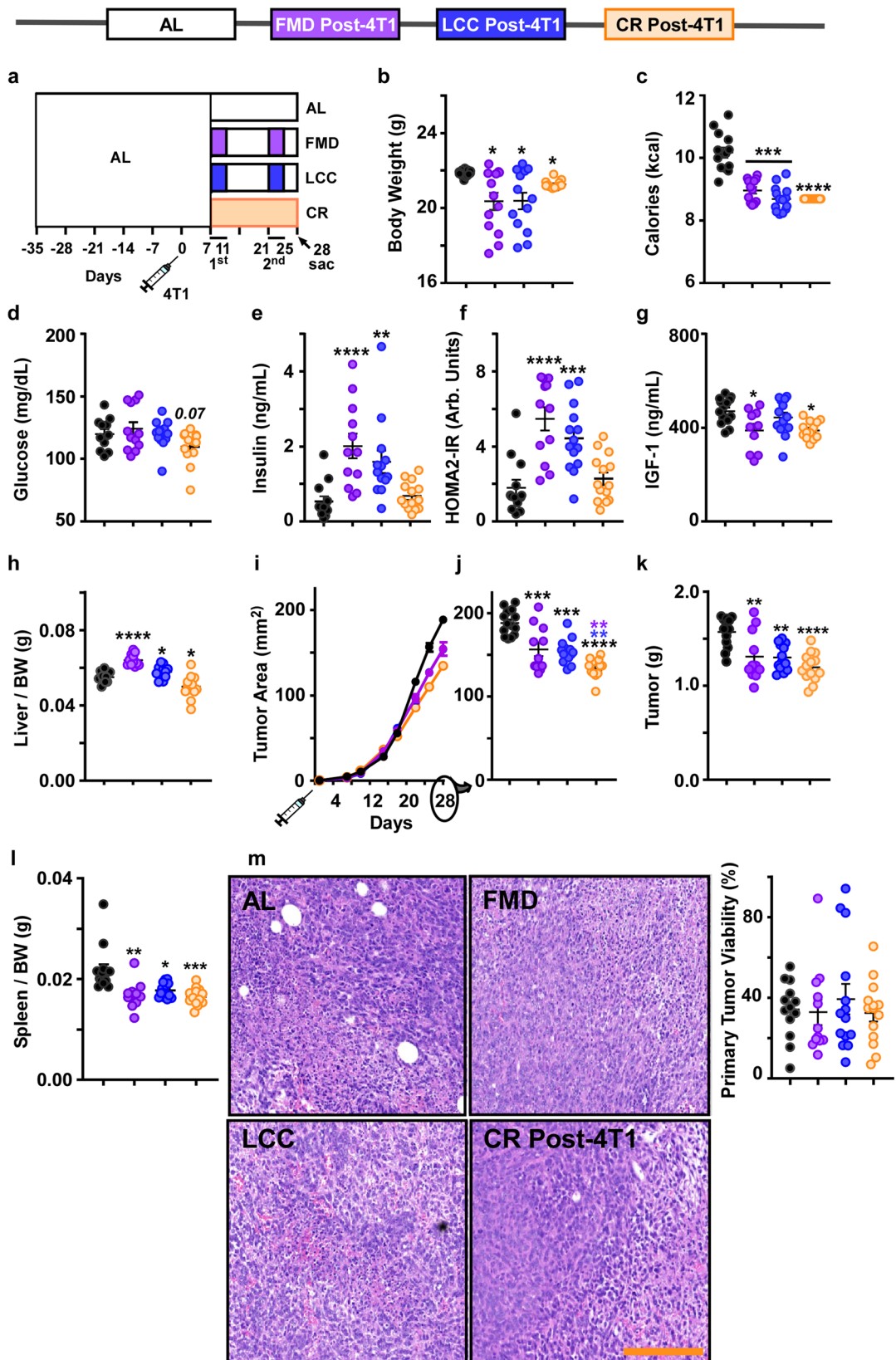

No substantial loss of lean or fat mass was observed with either CR or caloric cycling (Supplementary Fig. 1c–e).

Metabolic markers measured at the time of sacrifice (third day of refeeding on the fourth cycle of FMD/LCC and 49 days on CR) revealed a significant drop in glucose levels in the daily CR-fed mice (Fig. 4d). No differences in insulin levels and HOMA2-IR

index were observed between the treatment groups and AL control (Fig. 4e, f), although IGF-1 levels were significantly lower in FMD- and daily CR-fed mice (Fig. 4g). FMD regimen led to a marked increase in liver weight (Fig. 4h), while FMD and LCC increased kidney weight (Supplementary Fig. 4f). There results suggest that the caloric cycling mice (FMD and LCC) underwent

**Fig. 2 Daily CR provides greater protection against primary tumor growth than caloric cycling when initiated concurrently. a** Experimental Layout. A post-tumor implantation daily CR-fed group was incorporated into the study. All treatment groups were on ad libitum feeding during injection with 4T1 cells ($10^6$ cells/mL) at day 0. One week after injection (day 7), mice underwent 2 rounds of caloric cycling (FMD or LCC) or daily CR until tissue collection at day 28. **b** Average body weight and **c** average calorie intake throughout the study. **d** Blood glucose, **e** serum insulin levels, **f** the homeostatic model assessment calculation of insulin resistance (HOMA2-IR), and **g** serum IGF-1 levels collected at day 28. **h** Liver mass per unit of body weight. **i** Growth rates of the primary tumor. **j** Primary tumor area at day 28. **k** Tumor mass and **l** spleen mass per unit of body weight. **m** Representative images of H&E staining of primary tumors, including both viable and necrotic areas, under different experimental conditions [original magnification ×200] and histological quantification of tumor viability. Scale bar = 100 μm. Most of the data are represented as scatter plots with mean values ± SEM. One-way ANOVA with Tukey post hoc analysis was used to determine statistical significance with *$p < 0.05$, **$p < 0.01$, ***$p < 0.001$, ****$p < 0.0001$ compared to AL (black), FMD (purple), or LCC (blue). **b, c, h–m** AL $n = 13$; FMD, $n = 12$; LCC, $n = 13$; CR, $n = 13$. **d-f** AL, $n = 12$; FMD, $n = 12$; LCC, $n = 13$; CR, $n = 13$. **g** AL $n = 13$; FMD, $n = 10$; LCC, $n = 13$; CR, $n = 13$ mice per treatment group. Source data are provided as a Source Data file. BW body weight, arb. units arbitrary units.

metabolic 'priming' prior to 4T1 implantation, allowing sufficient time for metabolic adaptation to cycles of severe calorie restriction and AL refeeding.

Once again, all interventions resulted in tumor growth reduction when initiated prior to 4T1 implantation, especially in the daily CR-fed mice (Fig. 4i, j), where concurrent reduction in tumor mass (Fig. 4k) and spleen weight (Fig. 4l) was noted, the latter possibly due to tumor-mediated immune reprogramming. Daily CR was the only intervention capable at significantly reducing lung metastatic burden (Fig. 4m, n). These findings support the notion that daily CR—irrespective of the temporal initiation of the diet—was better at decreasing tumor growth rate and lowering metastatic burden than caloric cycling regimens.

**Effective reduction in cancer metastasis with CR after primary tumor resection.** Surgical resection of the 4T1 primary tumor is partially effective at preventing cancer metastasis in the lungs[25]. Fourteen days after initial 4T1 cell injection, the breast tumor was surgically excised and the effectiveness of daily CR or an additional 4:10 cycle of FMD or LCC at reducing lung metastatic burden was assessed and compared to uninterrupted 28 days of tumor growth (Supplementary Fig. 5a). Compared to AL-fed mice, daily CR and cycles of FMD or LCC were equally effective at reducing the average body weight and caloric intake, which were also measured at three time points, whether the tumor was resected or not (Supplementary Fig. 5b–d). As shown above, daily CR and cycles of FMD or LCC led to reduced tumor size and mass and smaller spleens after 28 days of tumor growth. As expected, tumors that were allowed to grow for 28 days were considerably larger than those resected at 14 days post-implantation (Supplementary Fig. 5e–g). Mice on daily CR and caloric cycling had smaller tumor size when excised after 14 days (Supplementary Fig. 5e, f), with significant reduction in tumor mass, in response to CR and FMD, and a downward trend in the LCC group (Supplementary Fig. 5g). The resection of the tumor at day 14 was accompanied by smaller spleen at the time of sacrifice on day 28 (Supplementary Fig. 5h).

Lung metastatic count was significantly lower in all mice with tumors excised after 14 days than those with continuous tumor burden. None of the dietary interventions promoted further reduction in the number of metastases in mice with excised tumors, although a trend was observed in the CR group. Once again, daily CR was the only intervention capable of limiting the formation of lung metastases in mice with 28 days of tumor burden (Supplementary Fig. 5i, j). These findings demonstrate that daily CR is better at suppressing lung metastases in mice than cycles of FMD or LCC.

**Delay extension in tumor growth with the degree of CR.** As the majority of breast cancer cases occur in post-menopausal women, we sought to compare the benefits of varying degrees

of CR (10–40%) vs. 4:10 cycles of FMD and LCC in delaying primary tumor growth (Fig. 5a). CR levels dose-dependently decreased the average body weight and food consumption in 11-month-old retired female breeders compared to AL-fed controls (Fig. 5b, c, Supplementary Fig. 6a, b). Rate of food consumption was also recorded (Supplementary Fig. 6c). 30% and 40% CR exhibited a significant reduction in blood glucose levels without changes in insulin levels and HOMA2-IR (Supplementary Fig. 6d–f). In contrast to young females, retired breeders on FMD or LCC showed similar insulin levels and HOMA2-IR as AL-fed controls (Supplementary Fig. 6e, f). IGF-1 levels were significantly lower only with 30% CR (Supplementary Fig. 6g). The unique metabolic signature in the older females vs. young virgin mice suggest that age and reproductive status rather than tumor burden per se may serve as greater determinants of metabolic homeostasis.

CR interventions (20–40%) resulted in a significant dose-dependent decrease in primary tumor growth rate (Fig. 5d, e) and reduction in tumor mass, with 20% CR approaching significance (Fig. 5f). Spleen weight was also significantly reduced with CR (Fig. 5g). Unlike younger females, retired breeders on FMD or LCC were not protected from 4T1 tumorigenesis (Fig. 5d, e). Histological information obtained from H&E-stained tumor sections revealed that none of the dietary interventions altered tumor viability (Fig. 5h, l, top panel), although intervention with 20–40% CR led to significantly lower mitotic count, a well-established marker of growth rate[27] (Fig. 5i, l, top panel). Thus, it appears that 20–40% CR, but not caloric cycling (LCC or FMD), can delay primary tumor growth in post-reproductive females.

Like in younger females, post-reproductive mice on 20–40% CR had a significantly lower number of metastases (Fig. 5j and Supplementary Fig. 6h), including advanced metastases (>1.5 mm) (Supplementary Fig. 6i), whereas the 10% CR and LCC groups showed a downward trend. Histological quantification of H&E-stained lung sections revealed a significant decline in the total metastatic area with increasing levels of CR (Fig. 5k, l, bottom panels), a finding consistent with the changes in metastatic burden observed in young mice, even though post-reproductive females showed greater lung metastatic area compared to their younger counterparts (Supplementary Fig. 6j).

**Therapeutic effect of increased degree of CR in tumor growth delay.** A limitation of the prior study was the temporal difference in the initiation of dietary interventions. Indeed, CR was initiated 4 weeks prior to the first cycle of FMD or LCC and before tumor implantation, thus enabling CR mice with a longer period for metabolic adaptation prior to tumor inoculation. To assess whether this adaptation period may have been responsible for the improved anti-tumorigenic potential of daily CR (Fig. 5), we conducted an additional study in which middle-aged females were subjected to 10–40% CR concurrently with the caloric cycling (FMD and LCC) interventions, 7 days post-4T1 cell

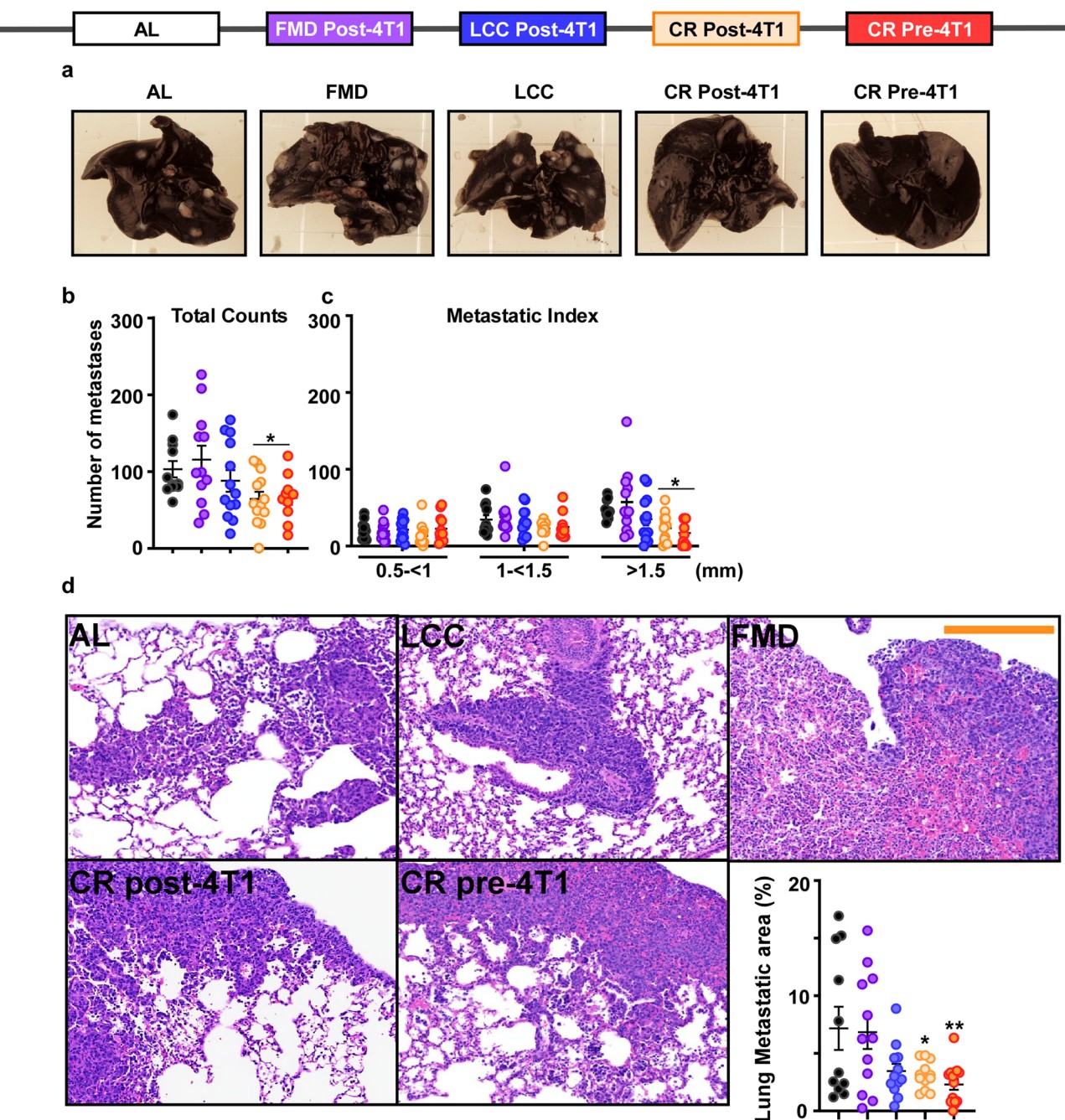

**Fig. 3 Preventive and therapeutic effect of CR but not caloric cycling in reducing metastatic burden. a** Representative images demonstrating the appearance of white masses in india ink-stained lungs, indicative of metastases. **b** Macroscopic quantification of lung metastases. **c** Macroscopically, lung metastases were scored in a blinded fashion and divided into three groups based on size: 0.5 < 1 mm, 1 < 1.15 mm, and >1.5 mm. Large primary tumors have a greater metastatic index. **d** Representative images and histological quantification of lung metastatic areas [H&E staining, original magnification ×200]. Scale bar = 200 µm. The lung and primary tumors presented in Fig. 2 and Supplementary Fig. 3 for each dietary intervention are from the same mouse. Most of the data are represented as scatter plots with mean values ± SEM. One-way ANOVA with Tukey post hoc analysis was used to determine statistical significance with *$p < 0.05$ compared to AL. **b**, **c** AL, $n = 11$; FMD post-4T1, $n = 12$; LCC post-4T1, $n = 13$; CR post-4T1, $n = 13$; CR pre-4T1, $n = 11$. **d** AL, $n = 11$; FMD post-4T1, $n = 12$; LCC post-4T1, $n = 12$; CR post-4T1, $n = 12$; CR pre-4T1, $n = 13$ mice per treatment group. Source data are provided as a Source Data file.

implantation (experimental design shown in Supplementary Fig. 7a). Increasing levels of CR led to a decrease in average body weight and at specific points during the experiment (Supplementary Fig. 7b, c). Caloric intake was also significantly lower during all dietary interventions compared to AL controls (Supplementary Fig. 7d).

Metabolic parameters were assessed at the time of sacrifice, after 21 days of feeding interventions. 40% CR significantly reduced the circulating levels of glucose (Supplementary Fig. 7e) and IGF-1 (Supplementary Fig. 7h), whereas none of the treatment groups had any impact on insulin levels and HOMA2-IR vs. AL controls (Supplementary Fig. 7f, g).

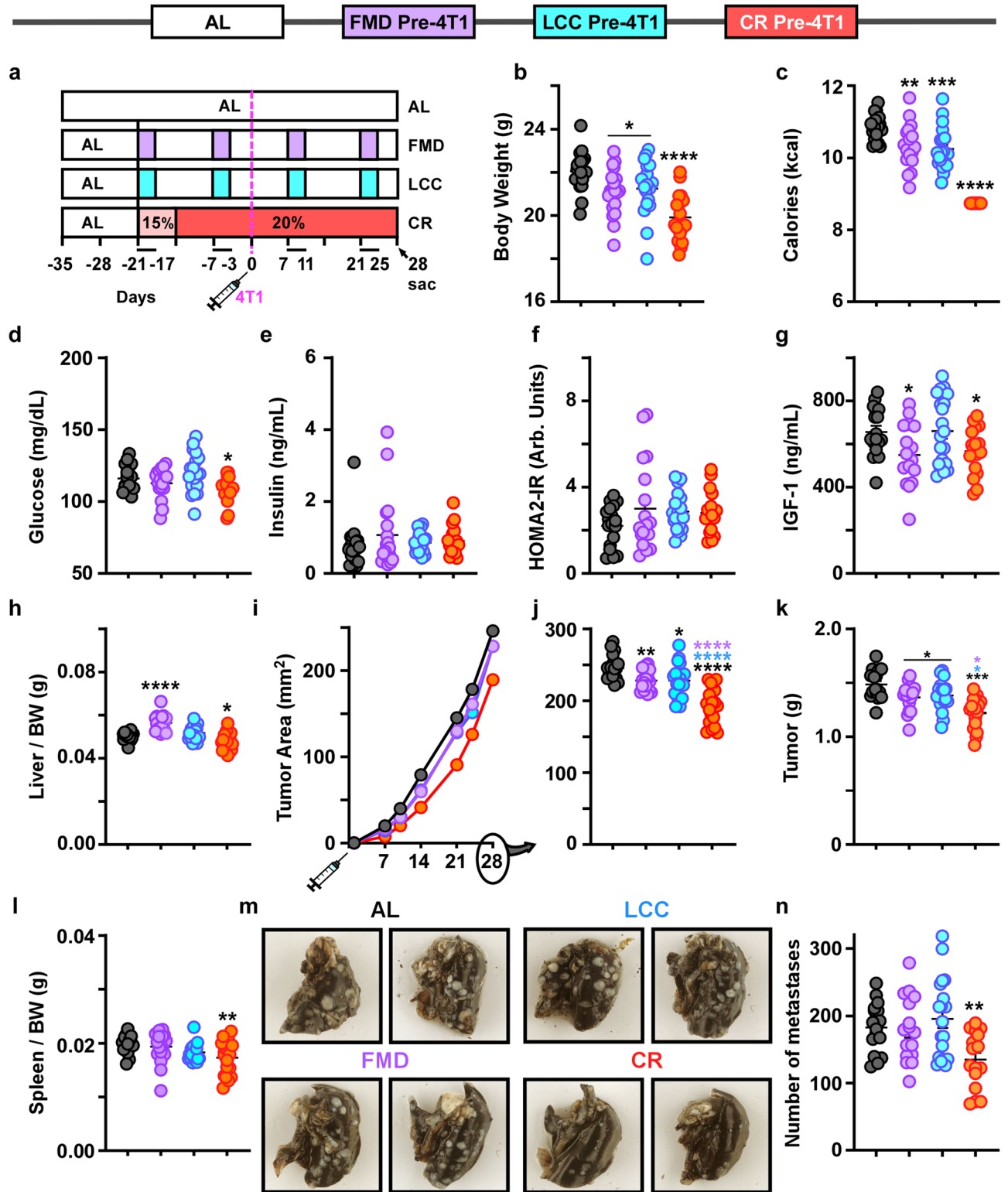

Therapeutic CR interventions (20–40%), like the prophylactic CR treatment, resulted in a significant, dose-dependent decrease in primary tumor growth rate (Supplementary Fig. 7i, j) and tumor mass (Supplementary Fig. 7k). Spleen weight was also significantly reduced in mice fed 20–40% CR vs. AL (Supplementary Fig. 7l). FMD and LCC did not protect against tumorigenesis (Supplementary Fig. 7i, j), in contrast to the protection conferred in younger females. Therapeutic CR (20–40%) also led to a significant reduction in the number of metastases (Supplementary Fig. 7m),

including advanced metastases (>1.5 mm) (Supplementary Fig. 7n, o). Taken together, these findings suggest that CR (20–40%) is effective at delaying primary tumor growth and metastatic burden, irrespective of whether it is initiated pre- or post-4T1 inoculation.

**CR and caloric cycling lead to unique immune signatures.**
Human and murine breast cancer growth and metastasis require regulatory immune cells, such as regulatory T cells (FoxP3[+]Tregs),

**Fig. 4 Initiation of caloric cycling and daily CR prior to 4T1 implantation slows primary tumor growth. a** Experimental layout. AL-fed BALB/cJ mice began the first cycle of caloric cycling (FMD and LCC) or daily CR concurrently, 21 days prior to 4T1 injection, or remained on AL. Daily CR underwent a stepwise decrease in caloric intake until 20% reduction was achieved. Two cycles of the FMD and LCC feeding regimen were completed prior to 4T1 implantation. **b** Average body weight and **c** average calorie intake throughout the study. **d** Blood glucose, **e** serum insulin levels, **f** the homeostatic model assessment calculation of insulin resistance (HOMA2-IR), and **g** serum IGF-1 levels collected at day 28. **h** Liver mass per unit of body weight. **i** primary tumor growth rate and **j** primary tumor area at day 28. **k** Tumor mass and **l** spleen mass per unit body weight. **m** Representative images demonstrating the appearance of white masses in india ink-stained lungs, indicative of metastases. **n** Total number of lung metastases. Most of the data are represented as scatter plots with mean values ± SEM. One-way ANOVA with Tukey post hoc analysis was used to determine statistical significance with *$p < 0.05$, **$p < 0.01$, ***$p < 0.001$, ****$p < 0.0001$ compared to AL (black), FMD (purple), LCC (blue). **b**, **c**, **h–j**, **l** AL, $n = 18$; FMD pre-4T1, $n = 17$; LCC pre-4T1, $n = 18$; CR pre-4T1, $n = 19$. **d–f** AL, $n = 18$; FMD pre-4T1, $n = 17$; LCC pre-4T1, $n = 18$; CR pre-4T1, $n = 16$. **g** AL, $n = 17$; FMD pre-4T1, $n = 16$; LCC pre-4T1, $n = 18$; CR pre-4T1, $n = 16$. **k** AL, $n = 18$; FMD pre-4T1, $n = 17$; LCC pre-4T1, $n = 18$; CR pre-4T1, $n = 18$. **n** AL, $n = 16$; FMD pre-4T1, $n = 16$; LCC pre-4T1, $n = 17$; CR pre-4T1, $n = 16$ mice per treatment group. Source data are provided as a Source Data file. BW, body weight; arb.units, arbitrary units.

B cells (Bregs), and myeloid-derived suppressor cells (MDSCs) to inhibit antitumor immune cells[28–32]. Therefore, we sought to determine whether daily CR and 4:10 cycles of FMD and LCC led to unique immune signatures.

Prior work has shown an upregulation of FoxP3$^+$CD4$^+$ Treg cells in murine models of breast cancer[31] and in patients with breast cancer[32]. FoxP3$^+$CD4$^+$ Treg cells promote lung metastases by inhibiting antitumor T cells and NK cells[28]. Here, we found a diet-dependent positive relationship between FoxP3$^+$CD4$^+$ Treg cells and primary tumor area or advanced metastases. Tumor-bearing mice on daily CR and cycles of FMD and LCC had a significant reduction in the frequency of FoxP3$^+$CD4$^+$ cells in peripheral blood vs. AL-fed controls (Fig. 6a, Supplementary Fig. 8a). FMD was the only regimen able to significantly reduce the percentage of splenic FoxP3$^+$CD4$^+$ Tregs (Fig. 6b, Supplementary Fig. 8b). Yet, in contrast to the CR-induced decrease in lung metastases (Fig. 3), metastatic progression was not hindered by LCC and FMD, suggesting the involvement of additional immune cells. Therefore, we quantified the frequency of FoxP3$^+$CD8$^+$ Tregs, which are reported to also be relevant in support of lung metastasis and progression in various murine cancer models, including prostate[33], colorectal[34], and breast cancer[35]. Although markedly reduced in the peripheral blood of LCC and CR mice (Fig. 6c, Supplementary Fig. 8c), there was only a significant decrease in FoxP3$^+$CD8$^+$ Treg levels in the spleen of CR mice (Fig. 6d, Supplementary Fig. 8d), suggesting that CR-mediated suppression of 4T1 lung metastases from the primary tumor may occur by inhibiting the activation and upregulation of these immunosuppressive cells.

Cancer also markedly expands myeloid suppressive cells such as MDSCs (as defined by the co-expression of CD11b and Gr1 surface markers) in the peripheral blood of tumor-bearing individuals[33–36]. In turn, MDSCs can directly or indirectly activate FoxP3$^+$CD4$^+$ Treg cells[37] and FoxP3$^+$CD8$^+$ Treg cells[33,35] and promote lung metastases. We did not detect any change in the frequency of MDSCs in the peripheral blood (Fig. 6e, Supplementary Fig. 8e); however, their frequency was significantly reduced in the spleen of CR animals (Fig. 6f and Supplementary Fig. 8f). A significant decrease in the ratio of FoxP3$^+$CD8$^+$ Tregs to MDSCs was observed in LCC and CR mice (Supplementary Fig. 9a).

MDSCs consist of two major subsets representing their granulocytic (PMN-MDSC) and monocytic (M-MDSC) origins. PMN-MDSC were identified by the co-expression of CD11b, Gr1, Ly6G, and low Ly6C, while M-MDSC were classified by the co-expression of CD11b, Gr1, and Ly6C). Both subsets increase in cancer and are implicated in poor breast cancer prognosis[38,39]. Interestingly, PMN-MDSCs were only significantly decreased in the peripheral blood of CR mice, while the peripheral blood and spleen of FMD and LCC mice saw an increase in their frequency (Supplementary Fig. 9b, c). In contrast to CR-fed mice, LCC

regimen led to significant increase in M-MDSC frequencies in blood (Supplementary Fig. 9d) and the spleen of the FMD and LCC groups (Supplementary Fig. 9e).

The loss of immune regulatory cells enables expansion of antitumor effector T cells (CD4$^+$ and CD8$^+$)[35,40,41]. The ratio between CD8$^+$ and CD4$^+$ cells to FoxP3$^+$CD8$^+$ Tregs is a key prognostic factor for different types of cancer. High ratios are associated with favorable outcomes in hepatic[42] and ovarian cancers[43]. Conversely, a shift towards a high number of Tregs is associated with poor prognosis in breast cancer[44]. Here, the ratio of T cells to MDSCs was markedly increased with daily CR and in response to FMD and LCC (Supplementary Fig. 9f, g), while only daily CR resulted in a significant rise in the ratio of cytotoxic T cells to FoxP3$^+$CD8$^+$ Tregs (Supplementary Fig. 9h, i). CR intervention also significantly increased the frequencies of tumor-cytotoxic effector T cells (Eff CD8$^+$ and CD4$^+$) in the spleen and peripheral blood (Fig. 6g–j and Supplementary Fig. 8g–j), including cytotoxic granzyme B (GrB$^+$) expressing effector T cells in the spleen, such as GrB$^+$CD8$^+$ and cytotoxic GrB$^+$CD4$^+$ (Fig. 6k, l, Supplementary Fig. 8k, l). Thus, the potency of CR in dampening tumor growth and metastasis could be explained by the inhibition of FoxP3$^+$CD8$^+$ Tregs, which, in turn, results in increased proliferation of antitumor cytotoxic T cells (CD8$^+$ and CD4$^+$) (Supplementary Fig. 9h, i).

Taken together, the use of a wide array of immune markers led us to uncover three predominant immunosuppressive markers in the peripheral blood and spleen that were significantly down-regulated in response to continuous CR.

**Remodeling of the tumor immune microenvironment.** The tumor microenvironment (TME) is one of the major epicenters for pro-tumorigenic immune reprogramming. As tumors favor an immune-suppressive environment, we sought to characterize changes in immune-dampening markers within the TME. Here, we found no significant differences in the amount of CD4$^+$Foxp3$^+$ Tregs (Fig. 7a, Supplementary Fig. 10a), matching prior findings in a 4T1 model subjected to 30% CR[45]. However, the frequency of CD8$^+$Foxp3$^+$ Tregs (whose levels are correlated with poor outcome[44]) showed a downward trend in daily CR-fed mice (Fig. 7b, Supplementary Fig. 10b).

We detected a significant reduction in total MDSCs in tumors from FMD and daily CR-fed mice (Fig. 7c, Supplementary Fig. 10c). Exploration of MDSC subsets revealed significant reduction in the frequency of PMN-MDSCs, but not M-MDSCs, only in CR mice (Fig. 7d, e, respectively). These findings match those found in tumors from breast cancer patients, wherein tumor infiltration is predominantly granulocytic MDSCs[46]. Tumor associated macrophages (TAMs), similar to MDSCs, promote tumor progression by blocking anti-tumor immunity and accelerating metastatic spread via angiogenesis[47]. Prior work

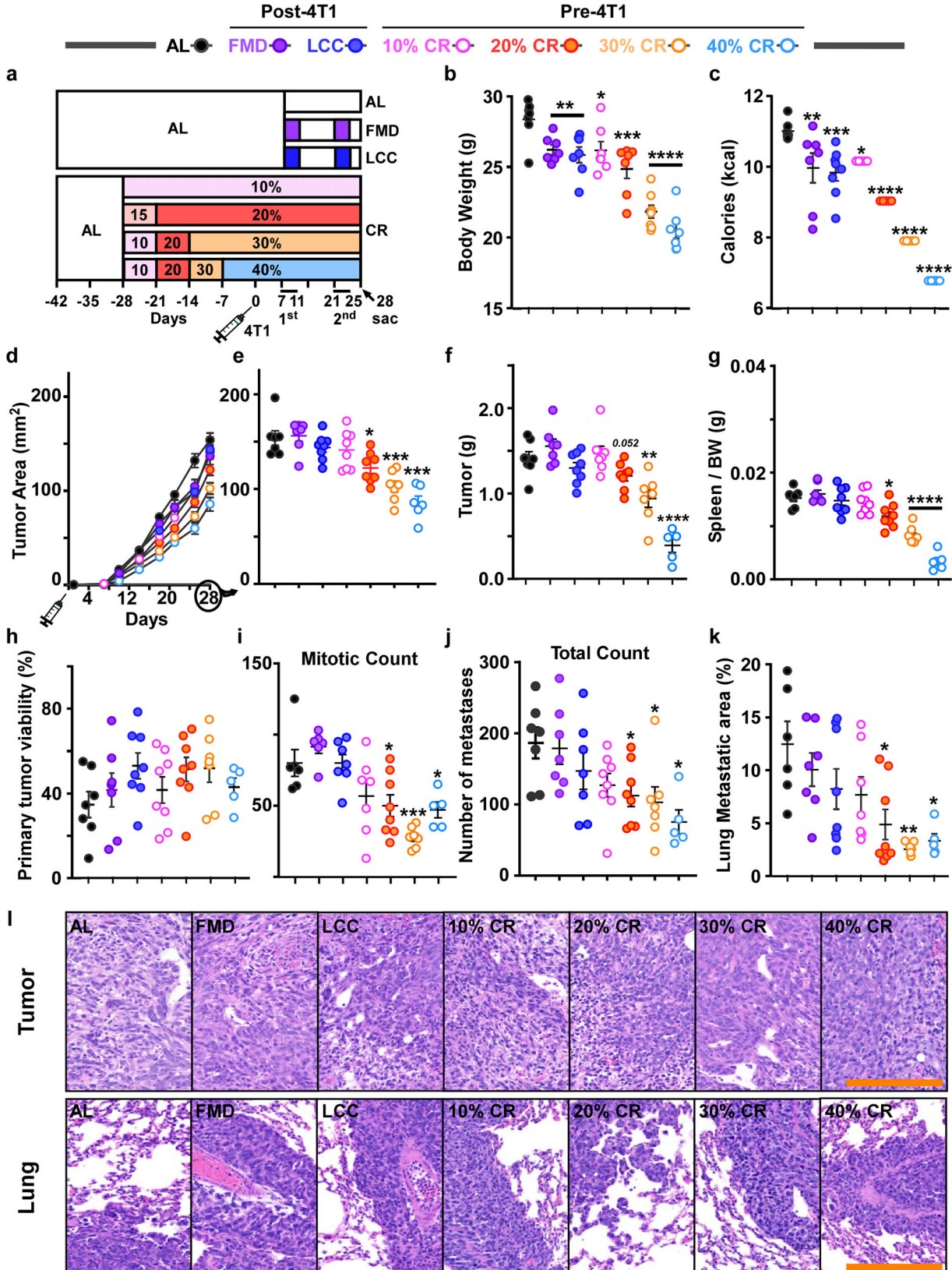

in the 4T1 model identified CD11b+CD163+ as the dominant immunosuppressive TAM[48], which was independently validated in TNBC biopsies[48,49]. A significant reduction in this marker was found only in CR-fed mice (Fig. 7f, Supplementary Fig. 10d).

Tumor-infiltrating lymphocytes (TILs) are predominantly comprised of CD8+ cytotoxic cells and CD4+ helper T cells,

two important factors for improved breast cancer survivorship[50]. Consistent with prior findings in the 4T1 cancer model[51], we did not find significant differences in CD4+ or CD8+ TILs levels within the TME (Fig. 7g, h, and Supplementary Fig. 10e), which may be indicative of the diverse array of mature and differentiated cells within this population[52].

**Fig. 5 Dose-dependent protective effect of daily CR against primary tumor growth in post-reproductive females. a** Experimental layout. AL-fed retired BALB/cJ breeders were randomized to varying degrees of CR (10–40%) using a ramp-down approach. Mice were then injected with 4T1 cells ($10^6$ cells/mL) at day 0 and remained on the specified doses of CR until tissue collection at day 28. Another AL-fed group of retired breeders was also injected with 4T1 cells and 7 days later was randomly divided into three groups; mice were maintained either on AL or subjected to two low caloric cycles of FMD or LCC before tissue collection at day 28. **b** Average body weight during the course of the study. **c** Average caloric intake. **d** Growth rates of the primary tumor. **e** Average tumor area at day 28. **f** Tumor mass and **g** spleen mass per unit of body weight (BW). **h** Histological quantification of primary tumor viability and **i** mitotic counts. **j** Macroscopic quantification of lung metastases [india ink-stained lungs]. **k** Histological quantification of lung metastatic areas. **l** Representative images of primary tumors (top) and corresponding lungs (bottom) under different experimental conditions [H&E staining, original magnification ×200]. Scale bar = 100 μm. Most of the data are represented as scatter plots with mean values ± SEM. One-way ANOVA with Tukey post hoc analysis was used to determine statistical significance with *$p < 0.05$, **$p < 0.01$, ***$p < 0.001$, ****$p < 0.0001$ compared to AL. **b**, **c** AL, $n = 7$; FMD, $n = 7$; LCC, $n = 8$; 10% CR, $n = 6$; 20% CR, $n = 7$; 30% CR, $n = 7$; 40% CR, $n = 6$. **d**, **e** AL, $n = 7$; FMD, $n = 7$; LCC, $n = 8$; 10% CR, $n = 8$; 20% CR, $n = 8$; 30% CR, $n = 7$; 40% CR, $n = 6$. **f–h**, **j** AL, $n = 7$; FMD, $n = 7$; LCC, $n = 8$; 10% CR, $n = 8$; 20% CR, $n = 8$; 30% CR, $n = 7$; 40% CR, $n = 5$. **i** AL, $n = 6$; FMD, $n = 6$; LCC, $n = 7$; 10% CR, $n = 7$; 20% CR, $n = 8$; 30% CR, $n = 7$; 40% CR, $n = 5$. **k** AL, $n = 6$; FMD, $n = 7$; LCC, $n = 8$; 10% CR, $n = 7$; 20% CR, $n = 8$; 30% CR, $n = 7$; 40% CR, $n = 5$ mice per treatment group. Source data are provided as a Source Data file.

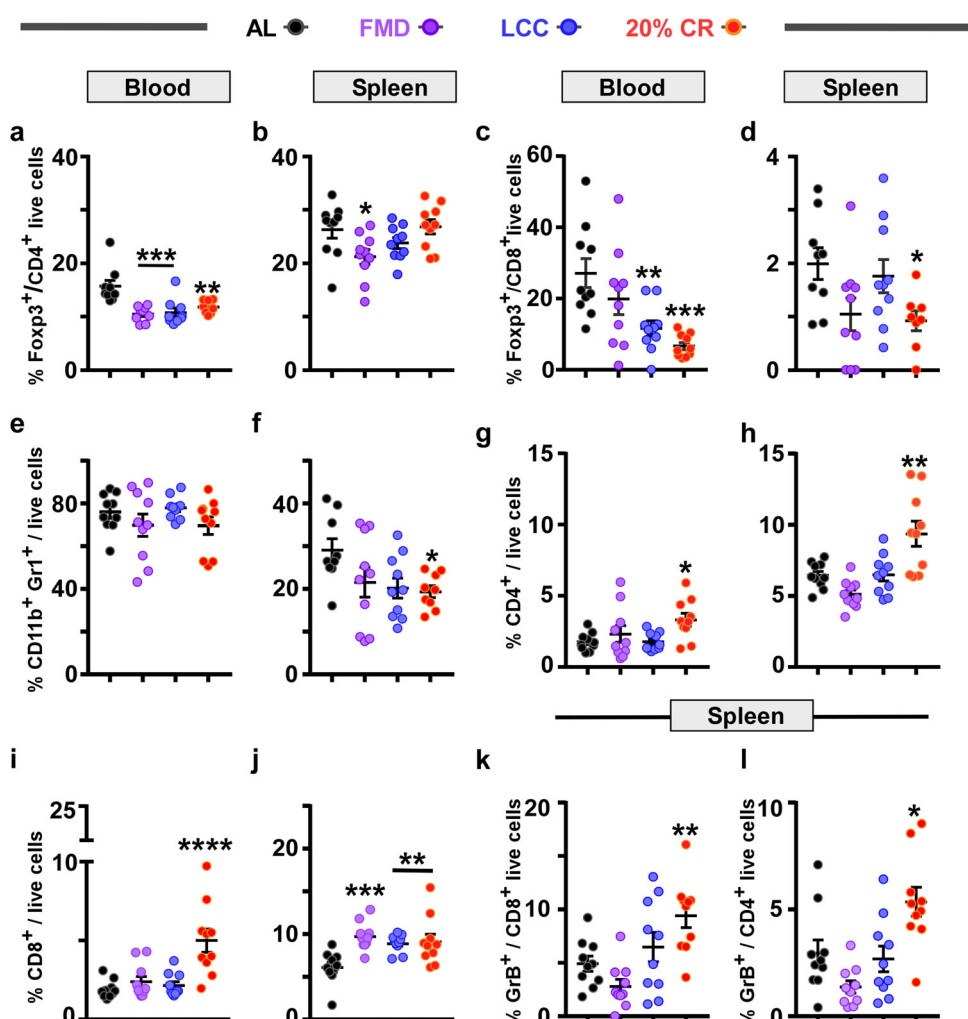

**Fig. 6 Daily CR leads to a unique immune profile in response to 4T1 tumor burden.** Percentage of immune cells detected in the peripheral blood (**a, c, e, g, i**) and spleen (**b, d, f, h, j**). **a, b** Foxp3+CD4+ T regulatory cells; **c, d** Foxp3+CD8+ T regulatory cells; **e, f** CD11b+Gr1+ cells (MDSCs); **g, h** effector CD4+ cells; **i, j** effector CD8+ cells. **k** Percentage of GrB+CD8+ cells and **l** GrB+CD4+ cells in the spleen. The data is represented as scatter plots showing mean values ± SEM. One-way ANOVA with Tukey post hoc analysis was used to determine statistical significance with *$p < 0.05$, **$p < 0.01$, ***$p < 0.001$, ****$p < 0.0001$ compared to AL. **a** AL, $n = 10$; FMD, $n = 9$; LCC, $n = 10$; CR, $n = 10$. **b, c, e, g-l** AL, $n = 10$; FMD, $n = 10$; LCC, $n = 10$; CR, $n = 10$. **d** AL, $n = 9$; FMD, $n = 10$; LCC, $n = 10$; CR, $n = 8$. **f** AL, $n = 9$; FMD, $n = 10$; LCC, $n = 10$; CR, $n = 9$ mice per treatment group. Source data are provided as a Source Data file.

The cytotoxic tumor-killing potential of TILs may be more indicative of delayed tumor progression. Indeed, a significant increase in Ly6C+-expressing memory T cells (CD4+Ly6C+) was found in the tumors of FMD and daily CR-fed mice (Fig. 7i and Supplementary Fig. 10f). In terms of CD8+Ly6C+ cells, increased

levels were evident in LCC and daily CR-fed mice (Fig. 7j, Supplementary Fig. 10g). CD103+ is a key marker in the identification of tissue-resident memory T cells that, by virtue of their enhanced cytotoxic killing ability, may protect tissues against disseminating tumors[53]. Increased expression of CD103+

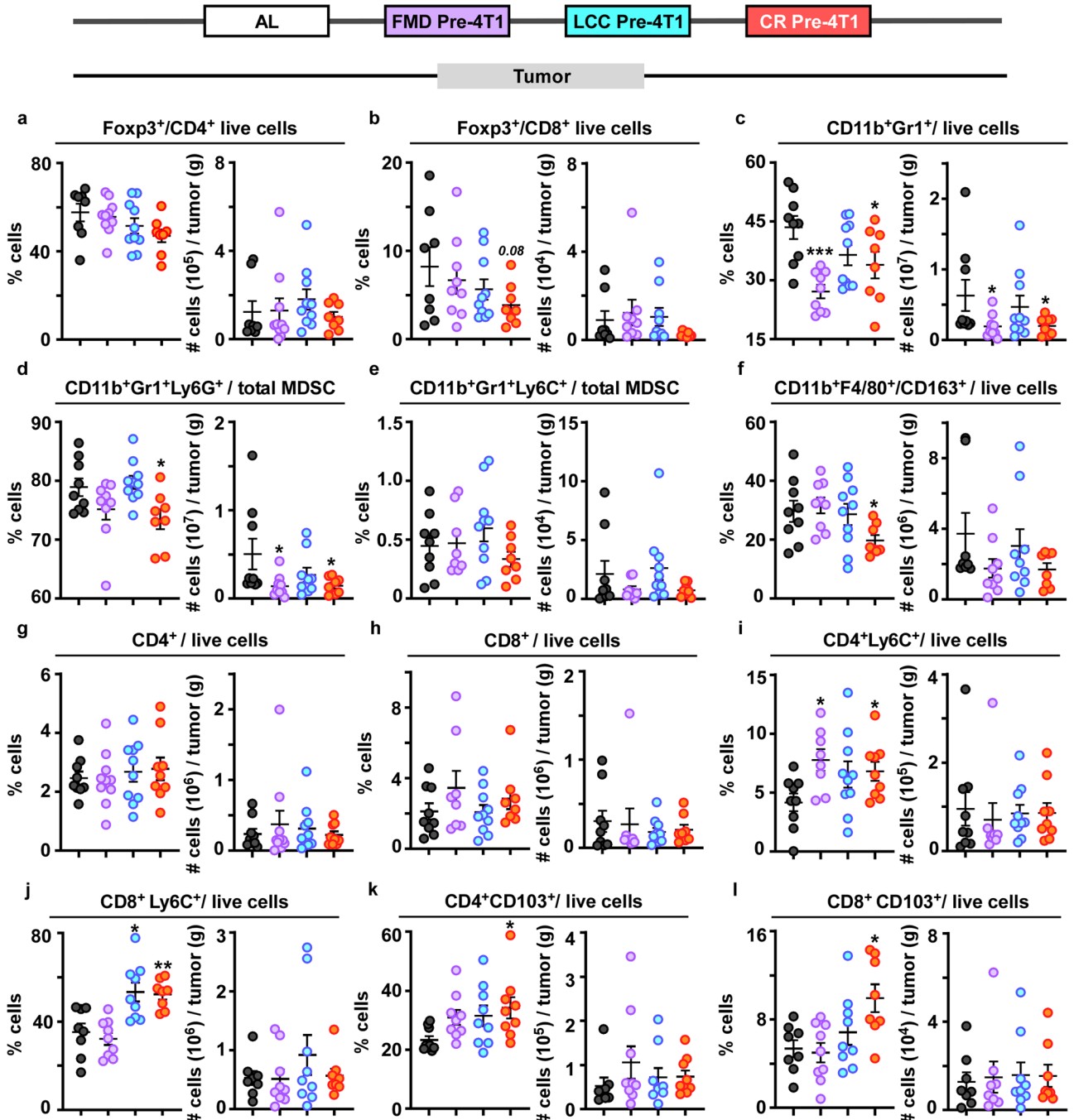

**Fig. 7 Immunological remodeling in the tumor microenvironment.** Frequency (%) (left) and cell count per gram of tumor (#) (right) of immune cells identified in the primary tumor collected from mice undergoing dietary regimens initiated prior to 4T1 implantation (Fig. 4). **a** Foxp3+CD4+ T regulatory cells; **b** Foxp3+CD8+ T regulatory cells; and **c** CD11b+Gr1+ cells (MDSCs). MDSC subset detected in the primary tumor, **d** CD11b+Gr1+Lys6G+ (granulocytic-MDSCs), and **e** CD11b+Gr1+Lys6C+ (monocytic-MDSCs). **f** CD11b+F480+CD163+ immunosuppressive cells. **g** Effector CD4+ and **h** effector CD8+ cells. **i** CD4+Lys6C+ cells and **j** CD8+Lys6C+ cells. **k** CD4+CD103+ cells and **l** CD8+CD103+ cells. Scatter plots represent mean values ± SEM. One-way ANOVA with Tukey post hoc analysis was used to determine statistical significance with *$p < 0.05$, **$p < 0.01$, ***$p < 0.001$. **a** AL, $n = 8$; FMD pre-4T1, $n = 10$; LCC pre-4T1, $n = 10$; CR pre-4T1, $n = 8$. **b** AL, $n = 8$; FMD pre-4T1, $n = 9$; LCC pre-4T1, $n = 10$; CR pre-4T1, $n = 8$. **c, d, f** AL, $n = 9$; FMD pre-4T1, $n = 9$; LCC pre-4T1, $n = 10$; CR pre-4T1, $n = 8$. **e** AL, $n = 9$; FMD pre-4T1, $n = 8$; LCC pre-4T1, $n = 10$; CR pre-4T1, $n = 8$. **g** AL, $n = 9$; FMD pre-4T1, $n = 10$; LCC pre-4T1, $n = 10$; CR pre-4T1, $n = 9$. **h** AL, $n = 9$; FMD pre-4T1, $n = 8$; LCC pre-4T1, $n = 10$; CR pre-4T1, $n = 8$. **i** AL, $n = 9$; FMD pre-4T1, $n = 8$; LCC pre-4T1, $n = 10$; CR pre-4T1, $n = 9$. **j, l** AL, $n = 8$; FMD pre-4T1, $n = 9$; LCC pre-4T1, $n = 9$; CR pre-4T1, $n = 8$. **k** AL, $n = 9$; FMD pre-4T1, $n = 9$; LCC pre-4T1, $n = 9$; CR pre-4T1, $n = 9$ mice per treatment group. Source data are provided as a Source Data file.

has also been linked with relapse-free survival in TNBC patients[54]. Noteworthily, higher CD103+ expression on CD4+ and CD8+ cells was evident only in daily CR-fed mice (Fig. 7k, j, and Supplementary Fig. 10h, i).

Histological assessment of TILs was not possible because of the morphological differences between human TNBC and the 4T1 murine tumor. Specifically, the gold standard for TIL assessment in human breast cancer[55] requires the evaluation of mononuclear

host immune cells only within the tumor stroma on H&E stained slides, as this has emerged as a robust prognostic and predictive biomarker[56]. Accordingly, in human TNBC, TILs are observed in the stroma between carcinoma cells and do not directly contact or infiltrate tumor cell nests (Supplementary Fig. 11a, b). In contrast, the 4T1 murine tumor is characterized by solid sheets of tumor cells with a high nuclear grade (Supplementary Fig. 11c) and lack intervening stroma, a well-known morphological feature of this mouse model[57]. The expansile invasive border was characterized by a very thin loose fibrous capsule populated by fibroblasts, mononuclear and polymorphonucleate immune cells. The tumor was often directly infiltrating the contiguous mammary adipose tissue. The adipocytes were surrounded by clusters of mononuclear and polymorphonucleate immune cells, which are not considered TILs (Supplementary Fig. 11d). Together these morphological differences prevented the histological evaluation of TILs.

## Discussion

CR is without a doubt the most robust non-pharmacological intervention against induced and spontaneous cancers. Numerous studies have shown the ability of daily CR to delay neoplasia in multiple tissues and inhibit the growth of chemically induced and spontaneous tumors[58], including breast cancer[59]. Despite the far-reaching improvements in health[60] and reduction in cancer incidence[59] associated with daily CR, its long-term implementation is not a feasible approach for most humans[61,62]. Implementation of protocols that involve IF as clinically viable alternatives to daily CR has been shown to promote similar improvements in metabolic markers[63], provide protection against cancer growth[10,17,64], and improve the response of a 4T1 murine cancer model to chemotherapy[21]. However, it is unclear whether optimization of diet composition in these less stringent feeding regimens would provide equal or better protection against tumor growth as daily CR. In this work, we begin to address this knowledge gap by directly assessing the effects of cycles of fasting-mimicking diet (FMD) vs. daily CR.

Periodic cycles of fasting/FMD followed by AL refeeding have been demonstrated to be a promising approach for treating cancer, both alone and in combination with either chemotherapy, kinase inhibitors, radiotherapy, hormone therapy, or immunotherapy[14–17]. Several clinical trials have reported successful outcomes[15,16,65]; however, important questions remain about the extent to which the composition of the FMD diet contributes to the observed beneficial effects and whether the effects of this form of caloric cycling can be as protective as daily CR against tumor growth and metastasis. Previous studies indicated that protein restriction, independent of calorie consumption, can delay the growth of certain tumors[6,19]. Here, we demonstrate that cycles of very low caloric intake either with AIN-93G chow (LCC) or a plant-based diet (FMD) were equally effective at delaying primary 4T1 tumor growth in young BALB/cJ females, a finding consistent with previous studies indicating that the growth of 4T1 breast cancer cells, contrary to other types of cancer[6], is not affected by protein restriction and the consequent lowering of IGF-1.

A key feature of this work is the demonstration that daily CR results in a more pronounced reduction in primary tumor growth than caloric cycling. This finding was consistently observed whether mice started CR and caloric cycling (LCC and FMD) pre- or post-4T1 implantation. A significant reduction in the tumor proliferative capacity (e.g., mitotic count) was achieved only in animals on increasing levels of daily CR (20–>40%). Therefore, it appears that daily CR and other less stringent forms of CR can limit tumor progression, a conclusion also reached by others[20,66].

Another key observation was the identification of immune remodeling in response to daily CR in peripheral tissues. Earlier work has suggested that upregulation of immune-fighting T-cells, specifically CD8[+] and CD4[+], leads to slower tumor growth[17] and is a good indicator of survival outcome[67]. Conversely, depletion of CD8[+] partially eliminates CR-mediated delay in tumor growth[45]. Here, our findings show that daily CR led to an increase in CD8[+] and CD4[+] cells in peripheral tissues. This is a key observation because T cell exhaustion is one of the major drivers of tumor progression and is indicative of poor prognosis in breast cancer patients[68]. Prior work found that nutrient-rich environments (i.e. a high-fat diet) led to accelerated T-cell exhaustion in a breast cancer model[69], whereas nutrient deprivation promotes apoptosis in malignant cells[10,12,23]. Within the TME of CR-fed mice, the number of CD4[+] or CD8[+] cells were not impacted, but their cytotoxic killing potential was elevated. Thus, CR may remodel the TME to dampen malignant cell growth while enhancing T cell proliferation and cytotoxic capacity. The capacity of daily CR to decrease the frequency of Foxp3[+]CD8[+], Foxp3[+]CD4[+], and MDSCs is consistent with protection from the metastatic potential of tumor cells[70–72]. These findings suggest that the immune phenotype elicited by daily CR may be exploited to target specific immune cells and provide opportunities for developing new therapeutic approaches. These alterations may contribute to the more robust cancer-fighting benefits observed with daily CR compared to caloric cycling.

Age is a major risk factor contributing to greater mortality in cancer patients, as post-menopausal women have a lower chance of survival than women in their peak reproductive years following breast cancer diagnosis[73]. Here, young female mice showed lower levels of metastatic lung tissue compared to post-reproductive females. This finding is important because it suggests that age and potentially hormonal shifts are key in survival outcomes. The diminished effect of caloric cycling (LCC or FMD) on tumor progression in older females constitutes another important finding of the study. Increasing levels of daily CR (20% or greater), irrespective if initiated pre- or post-4T1 injection, led to a significant reduction in tumor burden in older females, while the rate of tumor growth remained unabated in the caloric cycling groups. This may be in part because the reduction in caloric intake in response to caloric cycling in older females was less pronounced than that observed in younger animals. FMD regimen has been reported to reduce the negative side-effects of chemotherapy in post-menopausal women with HER2-negative stage II/III breast cancer, although the impact of FMD in the absence of chemotherapy could not be assessed on ethical grounds[15]. Further work is necessary to understand which of these interventions (daily versus cyclical calorie restriction) will be best suited to prevent and improve disease onset and outcomes in human studies.

The leading cause of mortality for breast cancer patients is the metastatic invasion to distal sites, including the lungs, rather than the primary tumor. Epidemiological data in breast cancer shows that metastases found in the lungs at diagnosis account for 60–70% of breast cancer-related deaths[24]. Daily CR has previously been shown to reduce lung metastases in mice[20,66]; however, less is known about the effectiveness of CR alternatives, including caloric cycling (LCC or FMD). Here, we demonstrated that even short-term daily CR, from 20% to 40%, was able to significantly reduce metastatic burden in both young and post-reproductive older females, whereas caloric cycling (LCC or FMD) was ineffective. Additionally, following excision of the primary tumor, caloric cycling (LCC or FMD) was unable to prevent lung metastasis progression, while CR showed a downward trend.

The FMD diet was originally designed to promote a synergistic effect in conjunction with chemotherapy. Numerous studies both

in animal models and human patients have indicated the potential for this approach to help deter cancer growth and reduce toxic side-effects when combined with chemotherapy or other therapeutic interventions[15,16]. Our findings suggest that when administered alone, daily CR (pre- or post-4T1 implantation) may promote a more robust anti-tumor response than caloric cycling. While adherence to daily CR regimens in long-term studies in humans has been a major issue, the short time frame of our studies and mild levels of CR (20%) that were needed to promote beneficial effects suggest that this could provide a viable therapeutic option for short-term intervention coupled with existing therapies. It will be extremely important for future studies to compare the implementation of daily CR versus caloric cycling in the context of chemotherapy.

A major concern regarding the use of CR in the clinical setting is fear of muscle loss or cachexia[74]. This is an important point to raise, especially in the context of the present study. The level of daily CR implemented for the majority of studies (20%) is mild and short-term (1–2 months). Even in mouse studies utilizing CR over the lifespan, decreases in body weight are mostly attributed to loss in fat mass and not lean body mass. Moreover, when normalized to body weight, mice subjected to continuous CR over the lifespan show no difference in lean mass compared to AL controls[3]. Indeed, in a murine cancer cachexia model, the implementation of daily CR (30%) two weeks prior to tumor inoculation has been reported to prevent loss of muscle strength, but not muscle mass, despite overall decrease in body weight[75]. Consistent with these findings, our results do not show a significant reduction in lean mass with daily CR or caloric cycling when initiated three weeks prior to tumor implantation.

Together, our findings demonstrate the ability of low caloric cycles to delay the growth of 4T1 breast cancer tumors regardless of diet composition. However, daily CR elicited a more robust reduction in tumor growth, whether the dietary interventions were initiated prior to or after tumor implantation. It would appear that older females were less responsive to the benefits of caloric cycling, and that daily CR was the only intervention capable of significantly reducing the incidence of lung metastases both in young and older females.

**Study limitations**. The main goal of the present study was to address the impact of calories, diet composition, and daily CR vs. LCC on tumorigenesis. Our follow-up work will evaluate how combining different diet compositions with standard therapeutic approaches for cancer treatment (e.g., chemotherapy) can influence tumor growth and metastases. The current findings are limited to a single tumor model (4T1 murine metastatic breast cancer). Therefore, additional work will be necessary to determine if similar outcomes occur in other types of tumors. It will also be important to evaluate whether caloric cycling can recapitulate the beneficial effects of daily CR on different cancer types, stages, and spontaneous tumor development in animals of both sexes.

## Methods

**Animals and diet**. Twelve-week-old BALB/cJ female mice and 36-week-old BALB/cJ retired breeders were purchased from Jackson Laboratories (Bar Harbor, ME). Mice were singly housed in duplexes (Thoren, #15 Single Housed Duplexed Cage; Dimensions 22.2 × 30.8 × 16.24 cm; Thoren Caging Systems, Hazelton, PA) from arrival with autoclaved corncob bedding and nestlets for enrichment at the NIA Biomedical Research Center (Baltimore, MD). Low-velocity HEPA filtered air was supplied through sealed shelf plenums directly into the cage through the air supply orifices above the cage filter top. Upon arrival and until one-week after tumor injection, all dietary regimens, except FMD, were fed AIN-93G Purified Rodent Diet (Dyets, diet #102423, Bethlehem, PA) for the entirety of the study. Water was provided AL in individual water bottles for each duplex. Baltimore City tap water treated by reverse osmosis and then hyper-chlorinated to 2–3 ppm was given. Cages were changed on a weekly basis in a biological safety cabinet, with spot changes as needed. Animal rooms were maintained at 22.2 ± 1 °C and 30–70%

humidity with 12-h day/light cycles. Upon study initiation, body weight and food consumption of ad-libitum-fed mice were measured twice weekly. In contrast, during the 4-day period of very low caloric intake, food consumption and body weight were determined daily. Mice with unrestricted access to the standard chow had the food placed in the hopper, while mice from the other feeding regimens had their pre-weighted daily allotment of food placed on the floor of the cage between 4:00 and 6:00 p.m. No significant amount of shredding was observed in any of the experimental groups. Mice were euthanized by $CO_2$ inhalation, followed by cardiac puncture. Serum and tissue were collected between 8 a.m. and 12 p.m., ~20 h after the last meal. All animal protocols were approved by the Institutional Animal Care and Use Committee (277-TGB-2024) of the NIA and performed under the Guide for the Care and Use of Laboratory Animals.

**Feeding regimens**

*Post-tumor caloric cycling (FMD and LCC)*. The fasting mimicking diet (FMD) is a plant-based diet designed to provide high nourishment during periods of low caloric intake[13,17]. Implementation of FMD is comprised of two parts, the first consists of a 4-day period of very low-calorie intake, followed by a 10-day refeeding period with unrestricted access to the AIN-93G chow (Supplementary Fig. 1). On the first day of the cycle, mice on FMD consumed only 50% of calories compared to AL-fed controls, and during days 2–4, they were further restricted, receiving only 30% of calories compared to AL-fed controls. The fasting standard diet (LCC) served as an isocaloric control, wherein mice were subjected to the same level of restriction and cycling as FMD-fed mice but were fed the AIN-93G chow. The first cycle began a week following 4T1 tumor cell injection with a subsequent cycle completed prior to euthanasia at day 28. Macronutrient composition is provided in Supplementary Table 1.

*Pre-tumor caloric cycling (FMD and LCC)*. Starting at 14 weeks, mice were maintained on the AIN-93G diet for 2 weeks, during which food consumption was recorded twice weekly for baseline feeding. Food consumption was averaged for the mice in the AL group and used to calculate the daily allotment for the 4-day period of very low-caloric intake. Two cycles of the low-caloric intake occurred prior to 4T1 implantation and were followed by two subsequent cycles completed prior to euthanasia at day 28.

*Pre-tumor daily CR*. Starting at 14- and 38-weeks, respectively, mice were maintained on the AIN-93G diet for 2 weeks, during which food consumption was measured twice weekly for baseline feeding. Food consumption was averaged for the mice in the AL group and used to calculate the allotment for daily CR. For CR ramp-down, a stepwise decrease in food intake was conducted each week until desired restriction was achieved and maintained for the duration of the study. It is important to note that the CR Pre-4T1 group was an additional treatment arm of the study presented in Fig. 2, and that the implantation and evaluation of tumors was performed concurrently with all five groups. Therefore, the same data for AL and CR post-4T1 groups are presented in Fig. 2 and Supplementary Fig. 3 (indicated by gray boxes), for the sake of clarity.

*Post-tumor daily CR*. The amount of food provided to CR Post-4T1 group was based on earlier studies performed in the same manner. Mice were given a daily allotment based on the average caloric intake across two cycles of FMD and LCC regimens. This amount was the same as the caloric intake observed in cycling groups in this experiment and resulted in approximately a 20% reduction in caloric intake similar to the daily CR group. Daily CR began one week after 4T1 cell injection, in conjunction with the first cycle of FMD and LCC.

**Cell culture and injection of 4T1 cancer cells**. Murine breast cancer 4T1 cells were purchased from ATCC (catalog # CRL-2539, Manassas, VA). Cells were free of *Mycoplasma*, as shown by a Mycoplasma Detection Kit (Lonza, Rockville, MD). They were cultured in complete RPMI 1640 medium supplemented with 2 mM L-glutamine, 10% FBS, 10 mM HEPES, pH 8, 1% sodium pyruvate, 1% nonessential amino acids solution, and 50 μg/ml penicillin–streptomycin (Gibco, catalog #11875101). Cells were grown at 37 °C in humidified 5% $CO_2$ incubators. Upon cellular confluency cells underwent trypsinization and were resuspended in sterile PBS and 100 μL were injected ($1 \times 10^5$ cells per mouse) subcutaneously in the fourth mammary gland on the right side of BALB/cJ female mice. Tumor size was measured with a digital vernier caliper.

**Resection of primary tumor**. Mice were anesthetized with isoflurane. Once mice were unconscious, the surgical area was shaved and sterilized according to NIH Animal Care Protocols. Tumors were removed by holding the tumor up with needle-nose forceps, enabling removal of the tumor away from healthy underlying tissue. Wounds were closed with Nexaband liquid (Henry Schein, Melville, NY). Afterwards, mice were monitored on a heated pad until they regained consciousness and placed into a fresh cage. Following surgery, mice were monitored daily under veterinary guidance. No recurrence of the primary tumor was evident.

**Glucose, insulin, and IGF-1 measurements**. Glucose was measured in whole blood using the Bayer Breeze2 handheld glucometer (Bayer, Mishawaka, IN)

immediately following euthanasia. Whole blood was collected post-euthanasia in prepared microtubes containing serum gel with clotting activator (VWR, catalog # 101093-958) and centrifuged at $8000 \times g$ for 6 min at 4 °C. Insulin and IGF-1 serum concentrations were determined using the Mouse Insulin ELISA kit (CrystalChem, Elk Grove Village, IL) and Mouse/Rat IGF-1 Quantikine ELISA kit (R&D Systems, Minneapolis, MN).

**Liver protein, triglyceride, and glycogen measurements**. Protein, triglyceride, and glycogen liver content was assessed following manufacturer instructions. Briefly, a piece of snap-frozen liver underwent dounce homogenization, using kit-specific buffer, before downstream processing. Lysate was analyzed for protein level (ThermoFisher Scientific, catalog #23225), triglycerides (Abcam, catalog #ab65336), and glycogen (Abcam, catalog #ab65620).

**HOMA calculation**. Insulin resistance was calculated using the blood glucose and insulin values that were entered into the HOMA2 Calculator software v. 2.2.2 available from the Oxford Centre for Diabetes, Endocrinology and Metabolism, Diabetes Trials Unit website (http://www.dtu.ox.ac.uk/).

**Flow cytometry**. Immune cells from spleen, blood, and primary tumor were collected and processed. Briefly, blood was collected in heparin-coated tubes, followed by incubation in ACK (Ammonium–Chloride–Potassium) lysis buffer for 10 min at room temperature (3× the volume of blood) (catalog #118-156-101, Quality Biological, Inc., Gaithersburg, MD). Afterwards, blood was centrifuged at $252 \times g$ for 5 min at 20 °C. The supernatant was removed, the pellet was resuspended in complete RPMI media (VWR, catalog #VWRL0106-0500), and then cell count was performed with final dilution at $4 \times 10^6$ cells/mL. Spleen was mechanically processed in a cell strainer (70 μm, Fisher Scientific, catalog # 07-201-431) and complete RPMI media was added. Tumors were mechanically minced and underwent enzymatic degradation, according to manufacturer instructions (Miltenyi Biotec, catalog #130-096-730). Processed tissue was centrifuged at $252 \times g$ for 5 min at 20 °C. The supernatant was removed, the pellet was resuspended in complete RPMI media, and cell count was performed, with final dilution at $4 \times 10^6$ cells/mL. Cells were plated onto a 96-well black plate (100 μL per well), and then stained following manufacturer's instructions (eBioscience, San Diego, CA).

The following flow cytometric antibodies and their isotype controls were used: CD11b BUV661 or BV785 (M1/70), Gr1 PE or Pacific blue (clone RB6-8C5), Ly6G PerCP Cy 5.5 (1A8), Ly6C BV510 (HK1.4), CD4 BV605 (GK1.5), CD8a BUV395 (clone 53-6.7), GrB PE (NGZB), FoxP3 PE (FJK-16s), CD103 PE (2E7), CD163 PE Cy7 (TNKUPJ), F4/80 APC (BM8) (Biolegend, Life technologies, eBioscience, or BD Biosciences, San Diego, CA) were used. Fixable Viability Dye eFluor™ 780 (eBioscience) was used for the staining of dead cells. All antibody concentration was used at 2 mg/mL with 1 μL antibody/$1 \times 10^5$ cells. Flow cytometry data were collected on a CytoFLEX platform (Beckman Coulter, CA) and analyzed with the CytExpert software v. 2.3 (Beckman Coulter) and FlowJo v. 10.8 (FlowJo, LLC).

**India ink staining**. India ink (10× solution of india ink in sterile PBS) was intra-tracheally injected. Lungs were fixed overnight in Feket's solution (300 mL 70% ethanol, 30 mL 37% formaldehyde, and 5 mL glacial acetic acid). Total number of white tumor nodules were counted against the black lung background. Additionally, each nodule was measured and separated by size ranging from 0.5 < 1 (small), 1 < 1.5 (medium), and >1.5 mm (large). Lungs were de-identified prior to counting.

**Histopathological assessment**. Tumors and lungs were collected 28 days after initial 4T1 injection. Tissues were incubated in 10% buffered formalin for 24 h, then transferred to 70% ethanol. Tissues were dehydrated in the following order: 70% ethanol for 1 h, 80% ethanol for 1 h, 90% ethanol for 1 h, 100% ethanol for 1 h, xylene for 2 h, and paraffin for 2 h. All tissues were processed for histology and H&E staining by Histoserv (Germantown, MD, USA). H&E-stained primary tumor and lung sections were scanned at ×200 magnification using the Zeiss Axio Scan Z1 digital scanner (Zeiss, Oberkochen, Germany). The acquired whole slide images (WSI) were evaluated with the open source bioimage analysis software QuPath v.0.2.2[76]. A board-certified pathologist annotated the de-identified WSI of primary tumor and lung sections. In primary tumor WSIs, all areas of viable tumor and the total tumoral area were manually annotated. As a result, the "tumor viability percentage" was calculated as a ratio of the total viable tumor area over the whole tumor area. In lung WSIs all tumor metastasis areas and whole lung area were annotated resulting in the "metastatic lung percentage" calculated as a ratio of the total metastatic area and the total lung area. Histological quantification of lung metastases was performed twice. First by a board-certified pathologist and then by two blinded investigators. The results were overlapping. Finally, mitotic count in primary tumors WSI was assessed by a board-certified pathologist by manually counting mitoses in the (visually selected) regions with the highest proliferative activity in a standard area of 2.37 mm[77]. Histologically, mitotic figures included nuclear aggregates lacking a nuclear membrane and with definite hyperchromatic hairy projections of nuclear material (chromosomes)[78]. Purely hyperchromatic or pyknotic nuclei were ignored. Chromosomal aggregates closely situated in two distinct clumps (telophase) were counted as one mitotic figure. Most of the slides were checked twice at two different times to account for intra-observer reproducibility.

Morphologically, the 4T1 cell-derived tumors were featured by a predominant solid and/or spindle cell pattern of growth, characterized by solid sheets of tumor cells with a high nuclear grade (Supplementary Fig. 11c). Regardless of the intervention and the age of the mouse, none of the tumors displayed any intervening stroma, which is a well-known morphological feature of all mouse models. This prevented the evaluation of TILs[55]. Accordingly, per recommendation of the "International Immuno-Oncology Biomarker Working Group (TILs-WG)", TILs should be assessed only in the tumor stroma on routine H&E-stained slides or slide images, as this standardized method has emerged as a robust prognostic and predictive biomarker in human TNBC[56]. A further spatial evaluation of possible stromal TIL-patterns was performed at the at the invasive border of the tumor (defined as the region centered on the border separating the host tissue from the tumor with an extent of 1 mm), based on the assumption that given the lack of stroma within the tumors, this would be a physical barrier for the TILs to enter. When assessable, the expansile invasive border was characterized by a very thin loose fibrous capsule populated by fibroblasts, mononuclear and polymorphonuclear immune cells. Often, the tumor was directly infiltrating the contiguous mammary adipose tissue, in which the adipocytes were surrounded by clusters of mononuclear and polymorphonucleate immune cells, which are not considered as TILs as such (Supplementary Fig. 11d). The limited number of stained sections with an invasive margin available for evaluation prevented the proper quantification and correlation of these patterns of infiltration with the interventions performed in the study.

**Statistical analysis**. Data is presented as mean ± standard error of the mean (SEM) unless otherwise specified. GraphPad Prism v8.4.2 (GraphPad Software, San Diego, CA) was used to generate graphs and statistical analysis of data. One-way ANOVA with Tukey post hoc analysis was used for comparisons between multiple groups, unless otherwise stated, with $p$ value of ≤0.05 considered statistically significant. Correlation analysis was conducted using Spearman correlation test with probability values of $p \le 0.05$ considered significant. Two-way ANOVA was also used (feeding regimen, tumor burden on days (14 vs. 28), and feeding × time interaction) followed by Tukey's multiple comparison test to determine significance between groups (Supplementary Fig. 5).

**Reporting summary**. Further information on research design is available in the Nature Research Reporting Summary linked to this article.

## Data availability
Resources, reagents, and microscopy images are available from the corresponding author upon reasonable request. The source data is provided within the article and as a supplementary file. Source data are provided with this paper.

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

## Acknowledgements
We are grateful to the Comparative Medicine Section of the NIA for their exceptional animal care. We thank Annamaria Rudderow and Jacqueline M. Moats for providing technical assistance in metastases quantification. We are also thankful to the NIA Visual Media Services, specifically Thomas Wynn, for providing photographic assistance in capturing the lung images. This study was supported by the Intramural Research Program of the NIA, NIH, and the Samuel Waxman Cancer Research Foundation. L.C.D.P.-W. was supported by the NIH Grant #Fi2GM123963 from the National Institute of General Medical Sciences of the National Institutes of Health.

## Author contributions
R.d.C., L.C.D.P.-W., M.B., and A.B. led the experimental design. R.d.C. and L.C.D.P.-W. supervised research experiments. L.C.D.P.-W., O.B., J.K., S.W., M.C., D.C., P.K., and S.N. completed animal work. O.B. and M.C. performed various biochemical and molecular analysis of serum. M.B., M.C., D.C., and E.R. completed immune cell isolation and staining. E.D., P.G.E., and R.S. performed the histopathological assessment of whole slide images. L.C.D.P.-W. and R.d.C. completed the data analysis. L.C.D.P.-W. wrote the original draft and created the figures. Interpretation, review, and final editing was performed by A.B., A.D.R., V.D.L., M.Ber., N.L.P., and R.d.C.

## Competing interests
The experimental FMD diet was provided by L-Nutra, Inc. The funding sources had no involvement in study design; collection, analysis, and interpretation of data; writing of the report; or decision to submit the article for publication. USC has licensed intellectual property to L-Nutra that is under study in this research. As part of this license agreement, the University has the potential to receive royalty payments from L-Nutra. V.D.L. who has equity interest in L-Nutra, did not participate in the collection and analysis of the data. One-hundred percent of V.D.L.'s equity will be assigned to the nonprofit foundation Create Cures. R.S. reports non-financial support from Merck and Bristol Myers Squibb, research support from Merck, Puma Biotechnology, and Roche and personal fees from Roche for an advisory board related to a trial-research project. He has no conflict of interest related to this work. The remaining authors have no competing interest to declare.
