## [Peer Review File · Nature Communications]

Daily Caloric Restriction Limits Tumor Growth More Effectively than Caloric Cycling, Regardless of Dietary CompositionREVIEWER COMMENTS

Reviewer #1 (Remarks to the Author); expert on diet intervention and cancer:

This study evaluates the effects of diet on breast tumor development and progression. The authors model triple negative breast cancer by implanting 4T1 cells into the breast mammary pad of young and old immunocompetent female mice. The objective is to compare the effects of daily fasting via calorie restriction (CR) with intermittent fasting using either the fasting mimicking diet (FMD) or a standard diet provided or the same duration of time as FMD (LCC). They use simple and effective readouts of systemic metabolism and tumor growth. They are studying an important topic and, overall, I like the concept, model system, and approach they are using. It would be of interest to the audience of Nature Communications. The authors find that the FMD and LCC inhibit cancer progression to the same extent. Pre-treatment with CR provided the greatest protection against the primary tumor growth and breast cancer metastasis to the lung. The authors attempt to connect these changes in tumor growth to systemic immunity but it is hard to interpret the relevance of these findings.

There is one major flaw in the study that must be addressed prior to publication. In several experiments, the authors compare mice fed FMD or LCC to mice fed CR. These groups are not directly comparable because the CR period begins prior to tumor implantation. This results in CR-treated mice with lower body weight (and altered metabolism) at tumor initiation being compared to normal weight mice that will be eventually be placed on FMD/LCC. In other words, the CR may be affecting tumor initiation and progression whereas the FMD/LCC treatment is only altering tumor progression. The best way to address this concern is to pre-treat the mice with pair-fed versions of the FMD/LCC and then implant the tumors when the body weight matches the CR-treated mice.

Specific Critiques:

Please describe how food consumption was measured? Was baseline food intake measured for each mouse and then implementation of a calorie deficit?

In prior publications there was 2 formulations of FMD: day 1 and day 2-4. Please define each dietary intervention used in the study by providing a table with the average energy intake per mouse (total kcal), the %kcal composition of macronutrients (carb, fat, pro, sugar, sat fat, unsat fat) and mass composition of the most common micronutrients (fiber, key vitamins).

If I understand correctly, it seems like the FMD has 2x more fat and 1/3 less carb than AIN-93G. FMD (CHO 44/ FAT 32/ PRO 18 %) vs AIN-93G (63/17/19 %)

Fig 1E suggests the mice fed LCC lost more weight during the CR cycles but is not shown in Fig 1F. In addition to average weight over the study, please show the weight at day 25 and at the end of the study in bar graph form.

Fig 1G has no error bars. Is the variation in the kcal intake from Fig 1H due to the variable eating during the 10 day adlib period?

The study in Figure 2 is flawed because CR preceded tumor implantation and FMD started after tumor implantation. The CR may be preventing the tumor from succeeding whereas the FMD is attempting to treat the tumor. The authors acknowledge this point on page 6, line 193-195, however, do not fully address my concern. The FMD and LCC should be implemented prior to tumor implantation until a similar amount of weight loss has occurred.

In addition to average body weight, please show the body weight at the time of tumor implantation and the body weight at the end of study

Figure 2H: is the increase in liver mass due to increased protein, triglyceride, or glycogen? These can be reported as mg/g liver.

The authors suggest that the reduction in spleen mass is unique to that organ. I suspect the reduction in organ mass occurs more diffusely. Was kidney, muscle, adipose tissue depots measured? Most likely all tissues have atrophied.

If the diet is altering tumor immunity as proposed, one would expect that the tumor microenvironment is altered. Please histologically assess the immune cell infiltration in the tumor.

In Figure 3, the authors pair-fed the dietary interventions at the start of tumor induction in order to control for the time differential in metabolic adaptation as compared to CR. This experiment was nicely controlled to determine the effects of ad lib vs pair-fed versions of FMD and LCC. However, it is inappropriate to compare these groups to the CR regimen, which was started prior to tumor injection and the mice (presumably) weighed less at time of tumor implantation. The effect of body weight at the time of tumor implantation is nicely demonstrated in Figure 4, where there is a dose-dependent effect of CR on tumor growth.

Please show the body weight at the time of tumor implantation and acknowledge these differences as a key limitation in the interpretation. If the authors insist on concluding that the CR leads to less tumor growth than the FMD/LCC regimens, then they should pair-feed FMD/LCC prior to implantation in order to match for the body weight at implantation.

Figure 3 – Total lung mass is a more robust measure of lung mets than “visible” mets or tumor area, which both can be biased by the observer.

Dose-dependent implementation of CR prior to tumor implantation in Figure 4. I liked this experiment because it clearly shows that more CR (and weight loss) leads to less tumor growth. I recommend removing this sentence: “Moreover, fasting time was found to be proportional to the level of CR, with animals maintained under the highest level of CR (40%) showing a fasting period of approximately 22 h” from page 8, line 246-248. It is an highly confounded association. The effect may be completely dependent on the weight loss with CR and have nothing to do with the fasting period. If the authors would like to test the effects of fasting intervals, then I recommend they use mouse cages with motorized food dispensers that allow access to small amounts of food at hourly intervals.

For the immune cell signatures presented in Figure 5, the daily and cyclical CR lead to unique immune signatures in the primary tumor vs. lung metastases. When were these cells collected? What was the body weight of the mice when the cells were collected. Given the previously mentioned limitations in study design (changes in body weight at the time of tumor implantation leading to less tumor burden), I do not know how to interpret this data. We may be seeing changes that are based on fat mass or other metabolic changes that are dependent on body weight.

Reviewer #2 (Remarks to the Author); expert on immunometabolism:

It is well known that caloric restriction can prevent tumor development, reduce tumor growth or improve efficacy of standard anti-tumor therapies. Pomatto-Watson and colleagues have performed a study to address the question what aspect(s) of caloric restriction (timing, duration, nutrient composition) is/are the main drivers behind this beneficial effect. They find that daily CR is superior to short cycles of plant or control diet-based CR in reducing tumor growth and development of metastasis in a murine model of breast cancer. Such a head to head comparison in the context of breast cancer has not been performed before and this work addresses an important issue, as it could help inform which type of CR may have the highest potential in limiting tumor growth or in improving chemotherapy. The manuscript is well written and overall conclusions are supported by the data. However, I do have some comments on some readouts and experimental approach that need to be addressed, before it would be suitable for publication.

Main comments:

1) In figure 2 the authors compare the effects of daily CR starting 2 weeks prior to tumor inoculation with cycling feeding of FMD and LCC starting 1 week after to tumor inoculation. This is a strange

comparison as the start of the intervention are 3 weeks apart of which the former is prophylactic and the latter is therapeutic. The authors do acknowledge this and partly address this issue in supplemental Fig 2 by comparing IF of FMD and LCC with daily CR of those diets during that same period. However, as the authors mention, the continuous FMD diet may not be suitable for this. So a better alternative for an appropriate comparison between daily CR and cycling IF FMD and LCC diets would be to start the latter two at the same time as the daily CR (day -14). Then duration of dietary intervention is the same and there will be no need for putting mice on continuous FMD diet that may have detrimental effect. Moreover, the authors should then start with this dataset and follow up with what is now shown in figure 2, to establish whether prophylactic vs therapeutic intervention has different effects in their model. That will be relevant for clinical translation.

2) The authors test in figure 4 the effects of varying degrees of CR on tumor growth in old mice. While they do not question the relevance of addressing this in old mice, there is a bit of a disconnect here with the preceding experimental setup as now 2 new variables are introduced: age and level of CR. Do the authors have data from young mice on varying degrees of CR, to be able to directly compare the difference in efficacy between these dietary interventions between young and old mice?

3) The immunological data from blood and spleen are interesting but lack information of the most relevant tissue in this context, the tumor itself. This is critical and should be added, both frequencies and cell numbers / g tumor tissue.

4) As indicated by the authors in the discussion it is important to know to what extent weight-loss associated with CR is a consequence of fat or lean (muscle) mass loss to be able to gauge for potential detrimental side effects. Do the authors have data on fat vs lean mass loss in the different dietary intervention arms?

5) the title should better reflect the findings of this study. 'Caloric restriction rather than dietary composition limits tumor growth' or something along those lines

Minor comments:

1) Please indicate in the first paragraph the age of the mice to make it clear that they are young mice as opposed to the mice used in Fig 4.

2) In the discussion it should be more clearly mentioned that the current findings are specific for breast cancer and that the relevance for other types of tumor remains to be determined

3) MDSCs come in neutrophilic and myeloid subsets. The flow data should include a Ly6C and Ly6G to be able to discriminate between the 2 of them

4) The authors are encouraged to discuss what their view is on how CR can lead to increased pro-inflammatory immune cell numbers to fight the tumor. Is this a direct effect on the immune cells themselves and/or a secondary effect of reduced tumor 'fitness' with its lower immune-suppressive capacity?

Reviewer #3 (Remarks to the Author); expert on metabolism and breast cancer:

Pomatto-Watson et al. present an interesting set of experiments to compare the effects of short term cycles of intermittent fasting compared to caloric restriction on tumor phenotypes in the 4T1 breast cancer model. They examine effects of the diets on tumor growth, spontaneous metastasis, and immune infiltrates in the tumors. They also evaluate how their findings might be affected when tumors are implanted in young versus older (post-breeding age) mice. In the end the authors conclude that the greatest effect is from ongoing caloric restriction. They also find that intermittent fasting has similar effects regardless of the diet tested, including so-called fasting-mimetic diets.

This is unquestionably a topic of interest, and the data should be published, but it was a struggle to follow exactly what was being done with some of the diet comparisons. The entire study is also based on analysis of one model, which does not necessarily invalidate the results, but does raise limitations in the ability to generalize conclusions. Suggestions that differences may be mediated in part by immune infiltrates are also highly speculative, and might be better suited as a starting point for a separate study.

1. It can be hard to follow some of the diet abbreviations. CR is standard, and FMD, AL, and IF are intuitive, but LCC for the low caloric cycling diet is very unclear and sorting through all the abbreviations when trying to follow the complex diet comparisons is challenging. This is problematic in the abstract, where upon first reading it is very hard to discern which diets are compared in the study.
2. Information on how the composition of the FMD diet compares to the control diet should be more readily provided.
3. The controls in the feeding experiments is not so clear. I assume in most cases they are ad-lib fed mice on a standard chow, but it is not always specified. It might also be appropriate to include an ad lib FMD control? I don't think this was done, unless it was part of the pair feeding data. Apart from being clear about what was done, it could be interesting if the FMD effect is further enhanced with reduced calorie feeding cycle.
4. The suggestion that shifts in immune cell populations account for the changes in tumor growth is quite speculative. While the findings are interesting, the story feels incomplete without testing this further at least in a cursory way. For example, are the effects of the diets lost in mice with compromised adaptive immunity? This seems unlikely given the many studies showing CR can work in xenograft models.
5. A major study limitation is that this entire manuscript is based on analysis of the transplanted 4T1 model, and thus rely on a single strain of mice. Differences in cancers or in whole body metabolism of different mice should at least be acknowledged as potential reasons these findings may not hold in all settings.
6. Are older "post-breeding" mice commonly used as a model of post-menopausal breast cancer? Is anything known about circulating hormone levels in these animals? If available citations for this should be included.

"Diet Composition or Calories: Their Impact on Cancer Growth and Metastasis"

REVIEWER COMMENTS

Reviewer #1 (Remarks to the Author); expert on diet intervention and cancer:

This study evaluates the effects of diet on breast tumor development and progression. The authors model triple negative breast cancer by implanting 4T1 cells into the breast mammary pad of young and old immunocompetent female mice. The objective is to compare the effects of daily fasting via calorie restriction (CR) with intermittent fasting using either the fasting mimicking diet (FMD) or a standard diet provided or the same duration of time as FMD (LCC). They use simple and effective readouts of systemic metabolism and tumor growth. They are studying an important topic and, overall, I like the concept, model system, and approach they are using. It would be of interest to the audience of Nature Communications. The authors find that the FMD and LCC inhibit cancer progression to the same extent. Pre-treatment with CR provided the greatest protection against the primary tumor growth and breast cancer metastasis to the lung. The authors attempt to connect these changes in tumor growth to systemic immunity, but it is hard to interpret the relevance of these findings.

There is one major flaw in the study that must be addressed prior to publication. In several experiments, the authors compare mice fed FMD or LCC to mice fed CR. These groups are not directly comparable because the CR period begins prior to tumor implantation. This results in CR-treated mice with lower body weight (and altered metabolism) at tumor initiation being compared to normal weight mice that will eventually be placed on FMD/LCC. In other words, the CR may be affecting tumor initiation and progression whereas the FMD/LCC treatment is only altering tumor progression. The best way to address this concern is to pretreat the mice with pair-fed versions of the FMD/LCC and then implant the tumors when the body weight matches the CR-treated mice.

Specific Critiques:

Q1. Please describe how food consumption was measured? Was baseline food intake measured for each mouse and then implementation of a calorie deficit?

R1.

Baseline food intake was measured for all mice prior to the initiation of any dietary intervention. Food consumption for mice in the ad libitum group was averaged and used to calculate the level of caloric restriction. The following was added to the Experimental Procedure section under 'Caloric Restriction Ramp-down' "Food consumption was averaged for the mice in the ad libitum group and used to calculate the daily allotment for daily CR."

Q2. In prior publications there was 2 formulations of FMD: day 1 and day 2-4. Please define each dietary intervention used in the study by providing a table with the average energy intake per mouse (total kcal), the %kcal composition of macronutrients (carb, fat, pro, sugar, sat fat, unsat fat) and mass composition of the most common micronutrients (fiber, key vitamins).

R2.

We have created an additional table (Table 1) that provides the dietary macronutrient composition for both the FMD and AIN-93G diets and cited this in the Experimental Procedure section. Micronutrient composition was unavailable. The average energy intake per mouse (total kcal) is provided in figure 1h,

figure 2c, supplemental figure 3d, figure 4c, supplemental figure 5d, Figure 5c, and supplemental Figure 7d.

Q3. Fig 1E suggests the mice fed LCC lost more weight during the CR cycles but is not shown in Fig 1F. In addition to average weight over the study, please show the weight at day 25 and at the end of the study in bar graph form.

R3.

For the study presented in Figure 1, we have added a new panel in supplemental figure 1b to depict bodyweight at the time of tumor injection (day 0), 25 days after tumor injection (day 25), and at the time of sac (day 28) when the tumor was collected.

Q4. Fig 1G has no error bars. Is the variation in the kcal intake from Fig 1H due to the variable eating during the 10-day ad lib period?

R4.

The revised Figure 1g now has error bars. The variation in the kcal in Fig 1h is due to the variable eating during the refeeding/ad libitum 10-day period.

Q5. The study in Figure 2 is flawed because CR preceded tumor implantation and FMD started after tumor implantation. The CR may be preventing the tumor from succeeding whereas the FMD is attempting to treat the tumor. The authors acknowledge this point on page 6, line 193-195, however, do not fully address my concern. The FMD and LCC should be implemented prior to tumor implantation until a similar amount of weight loss has occurred.

We apologize for any confusion resulting from the study depicted in Figure 2. The initial version of the manuscript contained data from one set of experiments divided between Figure 2 and Supplemental Figure 2 in an attempt to simplify our message. Although the study did include a group of animals subjected to CR that began at the same time FMD and LCC cycling was initiated (LCC-PF), we recognize that the nomenclature and organization of the data lacked clarity which led to considerable confusion. In the revised version of the manuscript, the data has been reorganized to facilitate the direct comparison between FMD/LCC cycling and CR initiated concurrently (7 days after implantation). The results clearly demonstrate that i) CR is more effective at reducing tumor growth and preventing metastases than FMD/LCC cycling even when all interventions are started at the same time, and ii) no differences in body weight are apparent at the time of tumor implantation. Supplemental Figure 3 further shows that pre-treatment with CR initiated two weeks prior to tumor implantation leads to a nearly identical effect to that observed when CR is initiated one week after implantation of the tumor.

As per the reviewer's recommendation, this set of studies has been repeated with all interventions initiated prior to tumor implantation, whereby two additional cycles of FMD and LCC were started along with daily CR prior to 4T1 implantation. The results demonstrate that CR was more efficient at delaying primary tumor growth and reducing lung metastases as compared to FMD/LCC cycling, when initiated prior to tumor implantation. See Figure 4 and Supplemental Figure 4.

Q6. In addition to average body weight, please show the body weight at the time of tumor implantation and the body weight at the end of study

R6.

Supplemental Figure 2a has been revised and now contains the bodyweight of each mouse at the time of tumor implantation (day 0), as well as at day 25 and day 28 (end of study). The text has been updated to reflect this new information.

Q7. Figure 2H: is the increase in liver mass due to increased protein, triglyceride, or glycogen? These can be reported as mg/g liver.

R7.

We have added data on liver protein, triglyceride, and glycogen into Supplemental Figure 2b-d and Supplemental Figure 3j-l. The following sentence has been added to the text: "Assessment of liver protein, triglyceride, and glycogen content showed only a significant decrease in total protein that was evident in all three dietary paradigms (Supplemental Figure 2b-d), suggesting changes in liver mass may be partially due to tumor-driven hematopoiesis."

Q8a. The authors suggest that the reduction in spleen mass is unique to that organ. I suspect the reduction in organ mass occurs more diffusely. Was kidney, muscle, adipose tissue depots measured? Most likely all tissues have atrophied.

R8a.

No significant decrease in kidney mass was observed in CR-fed mice vs. AL controls, indicating that not all tissues have atrophied under this intervention. See Supplemental Figure 4f.

Q8b. If the diet is altering tumor immunity as proposed, one would expect that the tumor microenvironment is altered. Please histologically assess the immune cell infiltration in the tumor.

R8b.

All H&E-stained primary tumor and lung sections from glass slides have been evaluated by three board certified pathologists (Eleonora Duregon, Roberto Salgado and Paula Gonzalez Ericsson), who were blinded to the experimental groups. R.S. and P.G.E. are world-renowned experts in immuno-oncology and the assessment of immune cells in breast cancer (PMID: 31086347, 32129476, 30894752), who are now co-authors on this manuscript. The International Immuno-Oncology Biomarker Working Group (TIL-WG), chaired by Roberto Salgado, recommends the assessment only of stromal tumor-infiltrating lymphocytes (TILs) on routine H&E-stained slides or slide images (PMID: 25214542), as this standardized method has emerged as a robust prognostic and predictive biomarker in human triple-negative breast cancer (PMID: 26667975). Stromal TILs are defined as mononuclear host immune cells (predominantly lymphocytes) observed within the tumor that are located within the stroma between carcinoma cells without directly contacting or infiltrating tumor cell nests (see new Supplemental Figure 12a and 12b). The tumors produced from 4T1 cells exhibit a predominant solid and/or spindle cell pattern of growth, characterized by solid sheets of tumor cells with a high nuclear grade (new Supplemental Figure 12c). Regardless of the intervention and the age of the mouse, none of the tumors displayed any intervening stroma, which is a well-known morphological feature in many mouse models (new Supplemental Figure 12c), making it impossible to apply the proposed method by TIL-WG. We also performed a spatial evaluation of possible stromal TIL patterns at the invasive border of the tumor (defined as the region centered on the border separating the host tissue from the tumor with an extent of 1 mm), based on the assumption that given the lack of stroma within the tumors, this would be a putative point of entry for TILs. The expansile invasive border was characterized by a very thin loose fibrous capsule populated by fibroblasts, mononuclear, and polymorphonucleate immune cells. Often, the tumor was directly infiltrating the contiguous mammary adipose tissue, in which the adipocytes were surrounded by

clusters of mononuclear and polymorphonucleate immune cells, which are not considered as TILs per se (new Supplemental Figure 12d). Due to the limited number of stained sections with an invasive margin, we were unable to properly quantify and correlate these patterns of infiltration with the interventions performed in the study. These observations have been included in the Results, Supplemental Figure 12a-d, and Experimental Procedure section.

Q9. In Figure 3, the authors pair-fed the dietary interventions at the start of tumor induction in order to control for the time differential in metabolic adaptation as compared to CR. This experiment was nicely controlled to determine the effects of ad lib vs pair-fed versions of FMD and LCC. However, it is inappropriate to compare these groups to the CR regimen, which was started prior to tumor injection and the mice (presumably) weighed less at time of tumor implantation. The effect of body weight at the time of tumor implantation is nicely demonstrated in Figure 4, where there is a dose-dependent effect of CR on tumor growth.

R9.

We apologize to the reviewer for the lack of clarity on our part. The description of each treatment group depicted in Figure 2 and Supplemental Figure 3 (formerly Figure 2 and Supplemental Figure 2) has been modified as followed: i) the FMD pair-fed group was removed because continuous feeding of the FMD diet was not sustainable for life, and ii) the LCC pair-fed group was renamed 'CR post-4T1'. Daily caloric intake for 'CR post-4T1' averaged a 20% reduction compared to the AL controls; this was calculated based on the average caloric intake that occurred between the ON and OFF cycles of the FMD and LCC groups. Noteworthy, mice enrolled in the 'CR post-4T1' arm of the study underwent daily CR initiated at the same time as the caloric cycling groups (FMD and LCC) and were fed the AIN-93G diet, as presented in Figure 2, Figure 3, and Supplemental Figure 2.

Additionally, the data associated with the CR group initiated prior to 4T1 implantation has been moved to Supplemental Figure 3 and this group was renamed 'CR pre-4T1'. Although both the 'CR post-4T1' and 'CR pre-4T1' arms were performed in the same study, the results were broken down into multiple figures in order to improve clarity as we describe the experimental design and results. Therefore, the AL and 'CR post-4T1' groups presented in Figure 2 were also used to compare to the 'CR pre-4T1' group shown in Supplemental Figure 3. The main conclusion from these experiments is that CR is equally effective of slowing primary tumor growth and lowering the number of lung metastases, irrespective of whether CR is initiated pre-tumor injection or post-tumor injection. CR offered a superior response than concurrent interventions with FMD / LCC cycles.

Q10. Please show the body weight at the time of tumor implantation and acknowledge these differences as a key limitation in the interpretation. If the authors insist on concluding that the CR leads to less tumor growth than the FMD/LCC regimens, then they should pair-feed FMD/LCC prior to implantation in order to match for the body weight at implantation.

R10.

The body weight at the time of tumor implantation has been added in Supplemental Figure 1b, Supplemental Figure 2a, and Supplemental Figure 3c. As suggested by the reviewer, we have incorporated new statements acknowledging the limitation of the original study and provided a better explanation and depiction of the data showing the superior response of CR vs. FMD/LCC cycles when these interventions were initiated after tumor implantation. We also performed an additional set of experiments in which all interventions began prior to tumor implantation.

Q11. Figure 3 – Total lung mass is a more robust measure of lung mets than “visible” mets or tumor area, which both can be biased by the observer.

R11.

We appreciate the reviewer’s comment; however, we felt that injection of the india ink stain introduces some degree of variation in lung weights, thus failing to provide an accurate mass. For each iteration, individuals counting the lung metastasis were blinded to the treatment group to prevent bias. The following statement has been added in the Experimental Procedure section under india ink staining. It reads: “Lungs were de-identified prior to counting.”

Additionally, a board-certified pathologist, who was blinded to the treatment groups, used H&E stained slides of lung sections to quantify lung metastases. The following statement has been incorporated in the Experimental Procedure section under tissue embedding & quantification. It reads: “A board-certified pathologist annotated the de-identified WSI of the primary tumor and lung sections.”

Q12. Dose-dependent implementation of CR prior to tumor implantation in Figure 4. I liked this experiment because it clearly shows that more CR (and weight loss) leads to less tumor growth. I recommend removing this sentence: “Moreover, fasting time was found to be proportional to the level of CR, with animals maintained under the highest level of CR (40%) showing a fasting period of approximately 22 h” from page 8, line 246-248. It is a highly confounded association. The effect may be completely dependent on the weight loss with CR and have nothing to do with the fasting period. If the authors would like to test the effects of fasting intervals, then I recommend they use mouse cages with motorized food dispensers that allow access to small amounts of food at hourly intervals.

R12.

We have removed this sentence and replaced it with ‘rate of food consumption was also recorded’.

Q13. For the immune cell signatures presented in Figure 5, the daily and cyclical CR lead to unique immune signatures in the primary tumor vs. lung metastases. When were these cells collected? What was the body weight of the mice when the cells were collected? . Given the previously mentioned limitations in study design (changes in body weight at the time of tumor implantation leading to less tumor burden), I do not know how to interpret this data. We may be seeing changes that are based on fat mass or other metabolic changes that are dependent on body weight.

R13.

Immunophenotyping was performed immediately following tissue collection at day 28 (end of the study). We have updated Supplemental figure 2a to show the individual body weights at the time of tumor injection (day 0), day 25, and day 28 (end of study).

Reviewer #2 (Remarks to the Author); expert on immunometabolism:

It is well known that caloric restriction can prevent tumor development, reduce tumor growth or improve efficacy of standard anti-tumor therapies. Pomatto-Watson and colleagues have performed a study to address the question what aspect(s) of caloric restriction (timing, duration, nutrient composition) is/are the main drivers behind this beneficial effect. They find that daily CR is superior to short cycles of plant or control diet-based CR in reducing tumor growth and development of metastasis in a murine model of breast cancer. Such a head-to-head comparison in the context of breast cancer has not been performed before and this work addresses an important issue, as it could help inform which type of CR may have the highest potential in limiting tumor growth or in improving chemotherapy. The manuscript is well written and overall conclusions are supported by the data. However, I do have some comments on some readouts and experimental approach that need to be addressed, before it would be suitable for publication.

Main comments:

Q1. In figure 2 the authors compare the effects of daily CR starting 2 weeks prior to tumor inoculation with cycling feeding of FMD and LCC starting 1 week after to tumor inoculation. This is a strange comparison as the start of the intervention are 3 weeks apart of which the former is prophylactic and the latter is therapeutic. The authors do acknowledge this and partly address this issue in supplemental Fig 2 by comparing IF of FMD and LCC with daily CR of those diets during that same period. However, as the authors mention, the continuous FMD diet may not be suitable for this. So, a better alternative for an appropriate comparison between daily CR and cycling IF FMD and LCC diets would be to start the latter two at the same time as the daily CR (day -14). Then duration of dietary intervention is the same and there will be no need for putting mice on continuous FMD diet that may have detrimental effect. Moreover, the authors should then start with this dataset and follow up with what is now shown in figure 2, to establish whether prophylactic vs therapeutic intervention has different effects in their model. That will be relevant for clinical translation.

R1.

We agree with the reviewer and have performed the proposed experiment. Two additional cycles of FMD and LCC were conducted prior to 4T1 cell inoculation and initiated concurrently with daily CR. Figure 4 and Supplemental Figure 4 illustrate the results. While tumor growth was delayed in all three interventions, daily CR was more efficient at impeding primary tumor growth and was the only intervention to reduce lung metastases significantly. The original data was also restructured to better compare FMD and LCC cycles with daily CR when all interventions were initiated after tumor inoculation.

Q2. The authors test in figure 4 the effects of varying degrees of CR on tumor growth in old mice. While I do not question the relevance of addressing this in old mice, there is a bit of a disconnect here with the preceding experimental setup as now 2 new variables are introduced: age and level of CR. Do the authors have data from young mice on varying degrees of CR, to be able to directly compare the difference in efficacy between these dietary interventions between young and old mice?

R2.

Unfortunately, we do not have data in young mice on different degrees of CR. However, prior studies using the 4T1-model in young BALB/cJ females show similar improvements in delayed tumor growth when mice were subjected to daily 10% (PMID: 31266827), 30% (PMID: 29912618), and 40% CR (PMID:

21665891). Nevertheless, one of our future aims is to assess the impact of varying levels of CR pre- and post-tumor implantation in other tumor models.

We also completed an additional set of experiments in which varying degrees of CR (10%-40%) were initiated in middle-aged females after 4T1 implantation, at the same time as caloric cycling (FMD & LCC). See new Supplemental Figure 7.

Q3. The immunological data from blood and spleen are interesting but lack information of the most relevant tissue in this context, the tumor itself. This is critical and should be added, both frequencies and cell numbers / g tumor tissue.

R3.

The data shown in Figure 4i-n were complemented by a new set of experiments where changes in tumor immune cells were measured and presented as frequency and cell number / g tumor tissue (Supplemental Figures 10 and 11). Like in peripheral tissues, there was a significant decrease in MDSCs and a downward trend in CD8⁺Foxp3⁺ T-regulatory cells, while no significant change in CD4⁺ and CD8⁺ cells was evident. However, markers of improved cytotoxic tumor killing ability were consistently elevated in daily CR-fed mice.

Q4. As indicated by the authors in the discussion it is important to know to what extent weight-loss associated with CR is a consequence of fat or lean (muscle) mass loss to the able to gauge for potential detrimental side effects. Do the authors have data on fat vs lean mass loss in the different dietary intervention arms?

R4.

In our most recent round of experiments, we have performed measurements of fat and lean mass prior to initiation of dietary intervention (baseline) and following two cycles of caloric cycling (FMD & LCC) or daily CR. This data, included in Supplemental Figure 4, shows that none of the treatments significantly reduce lean body mass with this type of short-term dietary intervention. We will closely monitor these outcomes in future studies, including those in which dietary interventions are coupled with chemotherapy.

Q5. the title should better reflect the findings of this study. 'Caloric restriction rather than dietary composition limits tumor growth' or something along those lines

R5.

We appreciate the suggestion and have changed the title as followed: 'Daily caloric restriction limits tumor growth more effectively than caloric cycling, regardless of dietary composition'

Minor comments:

Q6. Please indicate in the first paragraph the age of the mice to make it clear that they are young mice as opposed to the mice used in Fig 4.

R6.

The information regarding the age of the mice has been added in the first paragraph of the Results and in the Experimental Procedure section.

Q7. In the discussion is should be more clearly mentioned that the current findings specific for breast cancer and that the relevance for other types of tumor remains to be determined
R7.

The following statement has been added to the 'Study Limitations' section. It reads: "The current findings are limited to a single tumor model (4T1 murine metastatic breast cancer). Additional work will be necessary to determine if similar outcomes occur in other types of tumors."

Q8. MDSCs come in neutrophilic and myeloid subsets. The flow data should include a Ly6C and Ly6G to be able to discriminate between the 2 of them
R8.

The flow data now includes the monocytic-MDSCs (M-MDSC) and polymorphonuclear MDSCs (PMN-MDSC) subsets (Supplemental Figure 6). The results section now reflects this addition.

Q9. The authors are encouraged to discuss what their view is on how CR can lead to increased pro-inflammatory immune cell numbers to fight the tumor. Is this a direct effect on the immune cells themselves and/or a secondary effect of reduced tumor 'fitness' with a lower immune-suppressive capacity?
R9.

The following statement has been added to the Discussion to support a secondary effect of daily CR on tumor 'fitness'. It reads: "Here our findings show daily CR led to an increase in CD8⁺ and CD4⁺ cells. This is important because T cell exhaustion is one of the major drivers of tumor progression and indicates poor prognosis in breast cancer patients (78). Prior work found that nutrient-rich environments (e.g., a high-fat diet) lead to accelerated T-cell exhaustion in a breast cancer model (79), whereas nutrient deprivation promotes apoptosis in malignant cells (13, 18, 37). Thus, CR may remodel the tumor microenvironment to dampen malignant cell growth, while enhancing T cell proliferation and cytotoxic capacity."

Reviewer #3 (Remarks to the Author); expert on metabolism and breast cancer:

Pomatto-Watson et al. present an interesting set of experiments to compare the effects of short-term cycles of intermittent fasting compared to caloric restriction on tumor phenotypes in the 4T1 breast cancer model. They examine effects of the diets on tumor growth, spontaneous metastasis, and immune infiltrates in the tumors. They also evaluate how their finding might be affected when tumors are implanted in young versus older (post-breeding age) mice. In the end the authors conclude that the greatest effect is from ongoing caloric restriction. They also find that intermittent fasting has similar effects regardless of the diet tested, including so-called fasting-mimetic diets.

This is unquestionably a topic of interest, and the data should be published, but it was a struggle to follow exactly what was being done with some of the diet comparisons. The entire study is also based on analysis of one model, which does not necessarily invalidate the results, but does raise limitations in the ability to generalize conclusions. Suggestions that difference may be mediated in part by immune infiltrates are also highly speculative and might be better suited as a starting point for a separate study.

Q1. It can be hard to follow some of the diet abbreviations. CR is standard, and FMD, AL, and IF are intuitive, but LCC for the low caloric cycling diet is very unclear and sorting through all the abbreviations when trying to follow the complex diet comparisons is challenging. This is problematic in the abstract, where upon first reading it is very hard to discern which diets are compared in the study.

R1.

We apologize for the confusion. The term 'LCC' has been substituted by 'FMD's isocaloric diet equivalent' in the revised abstract. The statement reads as follows:

Using a breast cancer model in mice, we compared the potency of daily CR to that of periodic caloric cycling on FMD or an isocaloric standard diet against primary tumor growth and metastatic burden. Our findings show that FMD and its isocaloric equivalent equally inhibited cancer growth, while daily CR provided greater protection against the primary tumor growth and breast cancer metastasis to the lung.

The first paragraph of the Results section now contains the updated change. It reads:

[...]. We compared the responses of 4:10 cycles of FMD vs. an isocaloric standard laboratory chow (AIN-93G), which we have termed 'low caloric cycling diet' (LCC). [.....]

Q2. Information on how the composition of the FMD diet compares to the control diet should be more readily provided.

R2.

Supplemental Table 1 provides the macronutrient composition of the diet and is referenced in the Experimental Procedure section.

Q3. The controls in the feeding experiments are not so clear. I assume in most cases they are ad-lib fed mice on a standard chow, but it is not always specified. It might also be appropriate to include an ad lib FMD control? I don't think this was done, unless it was part of the pair feeding data. Apart from being clear about what was done, it could be interesting if the FMD effect is further enhanced with reduced calorie feeding cycle.

R3.

As stated in the figure legends, each study relies on ad libitum (AL)-fed mice as controls. We chose not to use an FMD-AL group as controls because this diet was not intended to be administered continuously and appears to be lacking essential nutrients needed to sustain life. Indeed, although isocalorically matched to the LCC-PF group (renamed 'CR post-4T1'), mice on FMD showed a rapid and substantial drop in body weight over the course of 14 days, which required us to provide half the calories from the AIN-93G diet in order to overcome the nutrient deficit and stabilize their body weight. In light of these facts, we concluded that the inclusion of an FMD-AL arm to the study would be ineffectual.

We agree with the reviewer about clamping the amount of food the caloric cycling mice receive during the refeeding period. It is a study we plan to complete in the future.

Q4. The suggestion that shifts in immune cell populations account for the changes in tumor growth is quite speculative. While the findings are interesting, the story feels incomplete without testing this further at least in a cursory way. For example, are the effects of the diets lost in mice with compromised adaptive immunity? This seems unlikely given the many studies showing CR can work in xenograft models.

R4.

The reviewer is correct that our data do not conclusively show that the alterations in immune cell populations are directly responsible for the reduced tumor growth observed in this study. While CR can reduce cancer risk in some models where adaptive immunity is compromised, CR may also promote immune remodeling (PMID: 33647370). However, the specific alterations through which CR exerts these effects remain largely unknown. We surmise that the immunomodulatory effects of CR in cell populations known to play an important role in tumor growth may also contribute to its anti-tumor protection. Particularly, alterations that occur with daily CR may contribute to its improved ability to combat tumor growth/metastasis as compared to caloric cycling. Future work is aimed at more directly demonstrating how these immunologic alterations contribute to the beneficial effects of CR.

Q5. A major study limitation is that this entire manuscript is based on analysis of the transplanted 4T1 model, and thus rely on a single strain of mice. Differences in cancers or in whole body metabolism of different mice should at least be acknowledged as potential reasons these findings may not hold in all settings.

R5

We agree with the reviewer that the use of one tumor model is a limitation of the study. Hence, the following statement has been added under the 'Study limitations' section. It reads: The current findings are limited to a single tumor model (4T1 murine metastatic breast cancer). Additional work will be necessary to determine if similar outcomes occur in other tumor types.

Q6. Are older "post-breeding" mice commonly used as a model of post-menopausal breast cancer? Is anything known about circulating hormone levels in these animals? If available, citations for this should be included.

R6.

Many of the pre-clinical studies in breast cancer utilize the 4T1-BALB/cJ model, typically performed in young female mice. Although most studies using retired breeders do not provide reasoning for their use, we surmised that these old females are most likely in estropause and experiencing reduced estrogen

levels. This outcome mirrors the timeframe typically recommended for hormonal therapy in women, which usually occurs early in menopause.

The following two citations have been added to the main text to address the age-dependent changes in hormonal levels in rodents, specifically at 9-12 months of age. These aged females experience irregular estrous cycles, dysregulated hypothalamic–pituitary–gonadal axis activity, and decreased response to luteinizing hormone release by estrogens.

Here Koebele, S.V., and Bimonte-Nelson, H.A. Modeling menopause: The utility of rodents in translational behavioral endocrinology research. *Maturitas* 87 (2016): 5-17.

Finch, C.E. The menopause and aging, a comparative perspective. *The Journal of steroid biochemistry and molecular biology* 142 (2014): 132-141.

REVIEWER COMMENTS

Reviewer #1 (Remarks to the Author):

Overall, the manuscript is much improved. Specifically, the data in Figure 1, 2, S1, S2, and S3 are more clearly presented and agree with the conclusions of the paper. I recommend to accept with minor revisions as follows:

Please describe HOW food consumption was measured. For example, were the food pellets pre-weighed, placed in the hopper, and then weighed again on a bi-weekly basis? Was a pre-weighed amount of food powder provided on the bottom of the cage and then the residual food was sifted out of the bedding on a bi-weekly basis? Or were the mice single housed in metabolic cages containing an electronic scale that measured the changes in food over time?

Please define each dietary intervention used in the study by providing a table with the energy density (kcal/g), the %kcal composition of macronutrients (carb, fat, pro, sugar, sat fat, unsat fat), and mass composition of the most common micronutrients (fiber, key vitamins). The provided table needs more information. The authors need to properly define the diets to ensure rigor and reproducibility.

- I also recommend adding comments such as "The AIN-93G diet contains 200 g of casein and 70 g of soybean oil/kg diet" and corresponding details for the FMD, which would complement the supplementary table. The information for AIN-93G can be found from the supplier or here <https://doi.org/10.1093/jn/127.5.838S>

- Please re-label the AIN column as "AIN-93G (LCC)" for clarity.

Please clarify the following sentence on page 17. "On the first day of the cycle, mice on FMD consumed only 50% of calories compared to AL-fed controls, and during days 2-4, they were further restricted (70% less calories)." Does this mean they consumed 50% of their normal calories on day 1 and then 30% of their normal calories on days 2-4?

Thank you for this contribution to the field.
M. Goncalves

Reviewer #2 (Remarks to the Author):

The authors have done an excellent job at addressing my concerns by significantly revising the manuscript and performing additional experiments that together have greatly helped to improve the quality of the manuscript and study. I have one final comment. Please move suppl fig 10 to the main figs, as these immunological tumor data are important and deserve a spot in the main body of the manuscript.

Reviewer #3 (Remarks to the Author):

The authors provided reasonable responses to my critiques, and their efforts to make the manuscript more clear are appreciated. With that said, the paper is not that different in content from the original, and I still think the immune infiltrate portions are quite speculative even though I am sure they will be of interest to many. Regardless, I stand by my assertion that this data will be of interest to many and should be published.

REVIEWERS' COMMENTS

Reviewer #1 (Remarks to the Author):

Overall, the manuscript is much improved. Specifically, the data in Figure 1, 2, S1, S2, and S3 are more clearly presented and agree with the conclusions of the paper. I recommend to accept with minor revisions as follows:

Q1. Please describe HOW food consumption was measured. For example, were the food pellets pre-weighed, placed in the hopper, and then weighed again on a bi-weekly basis? Was a pre-weighed amount of food powder provided on the bottom of the cage and then the residual food was sifted out of the bedding on a bi-weekly basis? Or were the mice single housed in metabolic cages containing an electronic scale that measured the changes in food over time?

R1. Mice were single housed and food was provided as pre-weighed pellets prior to daily feedings for the CR, LCC, and FMD groups. Ad libitum fed mice were provided a known amount of food placed in the hopper, which was weighed twice weekly, whereas the amount of food consumed by the CR and caloric cycling groups was determined daily. We did not notice a significant amount of shredding nor any other noticeable signs of distress. None of the mice were placed in metabolic cages.

Q2a. Please define each dietary intervention used in the study by providing a table with the energy density (kcal/g), the %kcal composition of macronutrients (carb, fat, pro, sugar, sat fat, unsat fat), and mass composition of the most common micronutrients (fiber, key vitamins). The provided table needs more information. The authors need to properly define the diets to ensure rigor and reproducibility.

R2a. The Supplemental Table 1 has been updated and now contains the requested information.

Supplemental Table 1: Macronutrient Composition.

Macronutrient composition of the AIN-93G diet and the plant-based, Fasting Mimicking Diet (FMD).

	AIN-93G (LCC)*		FMD	
Total energy density	3.76 kcal/g		2.92 kcal/g	
	g/kg diet	% kcal	g/kg diet	% kcal
Carbohydrate	601.0	63.9	320.3	44.5
Protein	177.0	18.8	62.0	8.6
Total fat (saturated)	72.0** (11.5)	17.2 (2.75)	150.1 (28.2)	46.9 (8.8)
Hydrogel	--		330.9	
Fiber	50.0	--	76.2	--
Vitamin and mineral mix (NR-1)			5.0	
Vitamin mix (#94047)	10.0			
Mineral mix (\$94046)	35.0			

* The AIN-93G diet contains 200 g of casein and 70 g of soybean oil/kg diet. Soybean oil has 16% saturated fat, 23% MUFA, and 58% PUFA. The AIN-93G diet also contains 2.5 g of choline bitartrate and 14 mg of TBHQ antioxidant/kg diet.

Q2b. I also recommend adding comments such as “The AIN-93G diet contains 200 g of casein and 70 g of soybean oil/kg diet” and corresponding details for the FMD, which would complement the supplementary table. The information for AIN-93G can be found from the supplier or here <https://doi.org/10.1093/jn/127.5.838S>

- Please re-label the AIN column as “AIN-93G (LCC)” for clarity.

R2b. The comments proposed by the reviewer have been added and the AIN column relabeled to AIN-93G (LCC). See R1b.

Q3. Please clarify the following sentence on page 17. “On the first day of the cycle, mice on FMD consumed only 50% of calories compared to AL-fed controls, and during days 2-4, they were further restricted (70% less calories).” Does this mean they consumed 50% of their normal calories on day 1 and then 30% of their normal calories on days 2-4?

R3. The reviewer is correct. This sentence has been modified as followed:

“On the first day of the cycle, mice on FMD consumed only 50% of calories compared to AL-fed controls, and during days 2-4, they were further restricted, receiving only 30% of calories compared to AL-fed controls.

Reviewer #2 (Remarks to the Author):

The authors have done an excellent job at addressing my concerns by significantly revising the manuscript and performing additional experiments that together have greatly helped to improve the quality of the manuscript and study. I have one final comment. Please move suppl fig 10 to the main figs, as these immunological tumor data are important and deserve a spot in the main body of the manuscript.

Thank you for your remarks. As per your recommendation, Supplemental figure 10 is now Figure 7.

Reviewer #3 (Remarks to the Author):

The authors provided reasonable responses to my critiques, and their efforts to make the manuscript more clear are appreciated. With that said, the paper is not that different in content from the original, and I still think the immune infiltrate portions are quite speculative even though I am sure they will be of interest to many. Regardless, I stand by my assertion that this data will be of interest to many and should be published.

We thank the reviewer for taking the time to review and provide valuable feedback to strengthen our manuscript.